# The Interplay between the Theories of Mode Coupling and of Percolation Transition in Attractive Colloidal Systems

**DOI:** 10.3390/ijms23105316

**Published:** 2022-05-10

**Authors:** Francesco Mallamace, Giuseppe Mensitieri, Martina Salzano de Luna, Paola Lanzafame, Georgia Papanikolaou, Domenico Mallamace

**Affiliations:** 1Istituto dei Sistemi Complessi, Consiglio Nazionale delle Ricerche, 00185 Rome, Italy; 2Department of Chemical, Materials and Production Engineering, University of Naples Federico II, Piazzale Tecchio 80, 80125 Naples, Italy; giuseppe.mensitieri@unina.it (G.M.); martina.salzanodeluna@unina.it (M.S.d.L.); 3Departments of ChiBioFarAm and MIFT—Section of Industrial Chemistry, University of Messina, CASPE-INSTM, V.le F. Stagno d’Alcontres 31, 98166 Messina, Italy; paola.lanzafame@unime.it (P.L.); georgia.papanikolaou@unime.it (G.P.); 4Departments of ChiBioFarAm—Section of Industrial Chemistry, University of Messina, CASPE-INSTM, V.le F. Stagno d’Alcontres 31, 98166 Messina, Italy; mallamaced@unime.it

**Keywords:** dynamical arrest, sol-gel transition, fragile-strong crossover, viscoelasticity

## Abstract

In the recent years a considerable effort has been devoted to foster the understanding of the basic mechanisms underlying the dynamical arrest that is involved in glass forming in supercooled liquids and in the sol-gel transition. The elucidation of the nature of such processes represents one of the most challenging unsolved problems in the field of material science. In this context, two important theories have contributed significantly to the interpretation of these phenomena: the Mode-Coupling theory (MCT) and the Percolation theory (PT). These theories are rooted on the two pillars of statistical physics, universality and scale laws, and their original formulations have been subsequently modified to account for the fundamental concepts of Energy Landscape (EL) and of the universality of the fragile to strong dynamical crossover (FSC). In this review, we discuss experimental and theoretical results, including Molecular Dynamics (MD) simulations, reported in the literature for colloidal and polymer systems displaying both glass and sol-gel transitions. Special focus is dedicated to the analysis of the interferences between these transitions and on the possible interplay between MCT and PT. By reviewing recent theoretical developments, we show that such interplay between sol-gel and glass transitions may be interpreted in terms of the extended F13 MCT model that describes these processes based on the presence of a glass-glass transition line terminating in an A3 cusp-like singularity (near which the logarithmic decay of the density correlator is observed). This transition line originates from the presence of two different amorphous structures, one generated by the inter-particle attraction and the other by the pure repulsion characteristic of hard spheres. We show here, combining literature results with some new results, that such a situation can be generated, and therefore experimentally studied, by considering colloidal-like particles interacting via a hard core plus an attractive square well potential. In the final part of this review, scaling laws associated both to MCT and PT are applied to describe, by means of these two theories, the specific viscoelastic properties of some systems.

## 1. Introduction

Scaling concepts, scale transformations and scale invariance have represented the conceptual background for the description of complexmaterials and system. Supramolecular materials (polymers, associating polymers, biological macromolecules, biological functions, colloids, amphiphiles, emulsions, etc.), aggregation processes, growth mechanisms, phase separation, out of equilibrium dynamics, supercooled liquids and glass transition represent examples of this topic overlapping research areas in physics, chemistry, biology, mathematics and technology.

Together with the scaling laws, the universality constitutes, according to the renormalization group theory, much of the correct representation of how these systems, made up of interacting subunits, behave [1]. It must be mentioned that these twin pillars were proposed by some pioneers exploring phase separation phenomena and the system behavior in the vicinity of its critical point.

Scaling also involve and explains, by means of the related correlation functions, both static and dynamic properties of the system in the stable as well as in the critical region. In this context, the mode coupling theory was used (MCT) [2], together with the classical hydrodynamic models [3,4]. From a more general point of view, the percolation theory and the MCT are the theoretical models that likely most exhaustively provide the structural and dynamic description of complex systems in terms of scaling laws.

Percolation theory (PT) is the simplest model displaying a phase transition. It basically describes the behavior of a network when nodes or links are added. It is a geometric type of phase transition, since at a critical fraction of addition, the network of small, disconnected clusters merges into significantly larger connected, so-called spanning cluster. The Flory–Stockmayer theory was the first model investigating percolation processes [5,6], and its application to materials science has become paramount to the understanding of many collective phenomena [7].

The MCT exploits the principles of the static scaling law in order to calculate the divergent part of the transport coefficients arising from the interaction among different modes of excitation of the system (e.g., viscosity, heat, sound and transport). Although initially developed to study dynamic critical phenomena [2,3,8] the MCT basic ideas can be of interest in a quite general context [9,10] (for example they play a central role for the liquid dynamics [11]). Here we consider both the theoretical models to explains the relevant and universal aspects of the dynamical arrest typical of liquids at the sol-gel transition and or supercooled ones on approaching the glass state [11,12,13]. As summarized by Phil Anderson in 1995, this phenomenon represents “*The deepest and most interesting unsolved problem in solid state theory is probably the theory of the nature of glass and the glass transition. This could be the next breakthrough in the coming decade*” [14]. As also predicted by him, and confirmed in recent years, the understanding of this process means that we can “*also have a substantial intellectual spin-off*” involving many scientific research fields.

However, in spite of both the theoretical and experimental interest placed on this topic in recent years, we are still far from an exhaustive understanding. In this context, it must also be mentioned that the experimentally observed huge variation with temperature (many orders of magnitude) of the shear viscosity, η(T), of supercooled liquids is the signature of slow dynamics, a characteristic phenomenon fundamental to the understanding of the glass transition.

A theoretical attempt to link the thermodynamic behavior of supercooled liquids to their flow and relaxation kinetics was due to the Adam and Gibbs analysis [15] developed by introducing the concept of elementary “cooperatively rearranging regions”. In such a way, an exponential expression connecting the temperature behavior of transport quantities (like the shear viscosity η(T), or alternatively a mean relaxation time τ(T)) to the configurational entropy per particle $S_{conf}$ was obtained. In the model, $S_{conf}$ is the portion of the entropy attributable to the “rugged landscape” character of the system’s potential energy function. In order to represent the vitrification kinetic aspects, two classes of scaling behaviors were empirically proposed when considering the η(T) evolution for several molecular glass forming liquid and polymers [16]: an Arrhenius (or strong) and a distinctly super-Arrhenius (or fragile) variation. In the same way, the glass transition temperature, $T_{g}$, is defined as the temperature at which η has the value of $10^{12}$ Pa.

The fragile scaling is the more intriguing of the two, as it indicates a thermal activation process where the activation barrier is T dependent, and the liquid heat capacities, Cp, are significantly larger than the corresponding crystal values (by an extent that increases with extent of supercooling). On the contrary, the strong scaling typically exhibits a little difference between liquid and crystal heat capacities. Nonetheless, the quest for understanding of metastable supercooling and glass formation has imposed an increasing level of attention. In any case nowadays, we have reached the certainty that the key to solving these problems lies in the analysis of the thermodynamic properties of transport of glass forming materials in both the supercooled and the glass states. For this reason, their energetic configuration on approaching the arrest as a function of the thermodynamic variables is of interest.

Accordingly, the idea of the “Energy Landscape and Inherent Structures” in the context of the several condensed-matter phenomena developed by Frank Stillinger is of utmost importance [17]. Exploiting these concepts, diverse substances and systems, including crystals, liquids, glasses and other amorphous solids, polymers, and solvent-suspended biomolecules can be explored. The topography of the multidimensional potential energy hyper-surface created when a large number of atoms or molecules simultaneously interact with one another is central in this approach.

The corresponding complex landscape topography separates uniquely into individual “basins”, each containing a local potential energy minimum or “inherent structure” [17]. On this basis, it has been demonstrated that just the energy landscape diversity is central for the study of the supercooled liquid properties on approaching the dynamic arrest [18]. It has been suggested that the potential-energy surface associated with Arrhenius (‘strong’) temperature dependence should be relatively smooth, whereas the energy surface giving rise to fragile behavior should be quite rough. An important finding of this study is also that the thermodynamics would appear to be decupled from kinetic behavior. Figure 1 elucidates the notion of the energy landscape in the description of supercooled liquids and amorphous materials [17].

In addition to these theoretical models on the explanation of the fragile behavior of supercooled glass forming liquids, many important suggestion have come from the use of atomistic concepts and from appropriate simulation studies [19]. They also provide a way to understand the nature of fragility in quantitative terms.

According to classical transition-state theory, η(T) of a glassy liquid can be expressed as
(1)$$\eta \left(T\right)=\eta_{0}\exp\left(Q\left(t\right)/k_{B}T\right)$$
in which $\eta_{0}$ is a reference value, Q(T) an effective activation barrier and $k_{B}$ the Boltzmann constant.

According to this, the understanding of the viscosity behavior relies on the explanation of the barrier and its details. The essential ingredient for this is the trajectory of the so-called transition-state pathway (TSP), an information obtained by sampling the potential energy landscape [20]. The TSP trajectory can be used for a direct viscosity calculation through linear response theory, without using the previous exponential form, by also elucidating our notion of the energy landscape description of amorphous media.

Many glass forming liquids appear to be “fragile” at high temperature whereas are “strong” at the lower temperatures. Thus, they change with *T* their energetic configuration implying a change in the corresponding landscape topographies—a situation that suggests, in these supercooled liquids, a possible connection between their thermodynamics and the kinetics. The onset of a sharp increase in η(T) over a relative small *T* range is characterized by a fragile to strong dynamic crossover (or critical) temperature $T_{x}$ (identified as the critical MCT $T_{c}$) [21]. This situation is characteristic of a very large number of glass forming liquids, and can be considered a universal process, providing further confirmation of the energy landscape model. In particular, the viscosity curves of these liquids collapse into a single curve only by assuming $T_{x}$ as the only characteristic temperature. Figure 2 illustrates this process by reporting the viscosity and diffusion data of more than 80 glass forming liquids. These data also show that below $T_{x}$, the familiar Stokes–Einstein relation ($D/T∼\eta^{-1}$) breaks down and is replaced by a fractional form $D/T∼\eta^{-\xi }$, with ξ≈0.85.

Such a result raises questions on which could be the change in the atomic interactions (and configurations) underlying such an upturn due to the temperature variation. In this regard, some calculations propose that the observed increase in the resistance to viscous flow can be interpreted as a molecular trapping in deep energy basins. In this energy framework, the crossover phenomenon—the viscosity transition from gradual to accelerated behavior—can be expected whenever different existing molecular mechanisms compete, with the dominance of one over the other depending on the local environment (or driving force). In this frame, an important analogy between viscous flow in glassy fluids and creep deformation in plastic solids was proposed [19].

Being that the glass rheology and crystal plasticity are closely related phenomena, the transition-state theory is equally suitable in describing stress-driven responses where Q(T) is replaced by a stress activation barrier Q(σ). Such an upturn is also a characteristic dynamic response experimentally observed in other systems from metals to colloidal suspensions [22], and may be compared to the η(T) variation with temperature of the fragile liquids. Such an upturn (or kink variation) is also a characteristic dynamic response experimentally observed in other systems from metals to colloidal suspensions [19]. Its wide spread presence proposes a common and simple underlying origin, basically a localization (or confinement) process. Thus, the η(T) crossover would correspond to stress localization as the system is trapped in a deep energy basin, whereas the crossover in the yield stress would correspond to strain localization when the strain rate reaches a critical value.

A common aspect of the recent year research in both MCT and PT was to readjust them in order to consider the physical effects related to the thermodynamic reality imposed by the energy landscape. In the first case, a semi-empirical version based on hopping processes [23,24,25] was proposed in addition to the idealized version [10]. On the other hand, for PT the idea of the random and the facilitated models was considered [26]. In such a way two important findings are obtained: (a) an interplay between gel and glass in gelling systems, with the possibility to describe the sol-gel transition using MCT, and simultaneously, to use percolation concepts in order to understand critical phenomena underlying the dynamic singularity found in MCT; and (b) the universal and singular role of the fragile to strong dynamic crossover on approaching the dynamic arrest. This was obtained starting from the consideration that both PT and MCT are characterized by very large viscosities (and relaxation times), but with basic differences.

The sol-gel transition (typical of the PT) has a diverging length and a strong divergence in the non-linear susceptibility (χ(T)). Instead, glass forming liquids, on approaching the arrest, do not show diverging length, critical density fluctuations and, therefore, no divergence in the compressibility. However, the energy landscape mechanism, if considered in both the models, predicts a special temperature $T_{x}>T_{g}$ where the energetic configuration changes. At this transition, being it made of highly correlated and localized clusters, the system can only have dynamics based on molecular hopping processes. As suggested by the Adam–Gibbs model, such a crossover temperature is the one where the “cooperative correlation length”, associated with a second-order thermodynamic transition, diverges. Such a length, associated with the internal degrees of freedom of the system, is unfortunately very difficult to verify experimentally. Nonetheless, the violation of the Stokes–Einstein relation (SE) and the onset of the dynamic heterogeneities is customary observed on $T_{x}$ in supercooled liquids [27,28].

The plan of this review is to consider the possible interplay between MCT and PT also based on the universality of the FSC and the relevance of the energy landscape concept to condensed matter and statistical physics. In such a context, we will discuss already published data in the field and present new ones, in the belief that the key to clarifying the complexities of dynamic arrest is precisely in the FSC, due to the change in the system energetic configuration.

## 2. Schematic Models

Many systems and models exhibit a sudden slowing down of their dynamics, followed by transition associated with a structural arrest. We can distinguish two types of transitions, continuous and discontinuous, depending whether or not there is a dynamic correlator jump in the infinite time limit. The first one is given by the sol-gel transition (commonly studied using a cluster approach, based on percolation theory) [26,29,30]. The glass transition instead, given by discontinuous transition, belongs to the second category. A great advance in glass theory was provided by MCT developed by Götze and collaborators [10,31].

Starting from first principles, under some mean field approximations, MCT predicts a dynamic arrest at a the finite $T_{c}$, characterized by power law behavior and universal scaling laws. These theoretical findings have been tested in great detail both experimentally and numerically [32,33,34,35,36,37,38]. Other models, such as p-spin glass, Random Field Ising in an external field [39] and kinetic facilitated [40], reproduce the same MCT results. However, the transition described by MCT does not seem to exhibit any critical change in the structure and no diverging static length.

### 2.1. Ideal MCT

The MCT [10,31] aims to capture the slow dynamics induced by slow relaxation of density fluctuations in dense liquids [10,12,13]. It formulates, for a system of *N* particles (with mass *m* and an average density ρ), a closed equation of motion for the relative time structural changes described by the coherent density auto-correlation function, where the correlator is:(2)$$\phi_{q}\left(t\right)=\langle \rho_{q}^{*}\left(q,t\right)e^{iŁt}\rho_{q}\left(q,0\right)\rangle /\langle |\rho_{q}{|}^{2}\rangle$$
where Ł is the Liouville operator and $\langle |\rho_{q}{|}^{2}\rangle /N=S_{q}$ (the static structure factor, and $\vec{q}$ the wave vector). The ideal MCT equations are a set of nonlinear coupled integro-differential equations that determine the dynamics of $\phi_{q}\left(t\right)$. Essentially, the static structure factor S(q) and similar equilibrium static correlation functions enter here. This allows for a first-principles comparison of the theory with simple glass-forming liquids, most prominently the hard- or soft-sphere system [41]. In these terms the idealized-MCT equations consist of the exact Zwanzig–Mori equation (ZM) [42,43] for a correlation-function description of the dynamics of a many-particle system:(3)$$\partial_{t}^{2}\phi_{q}\left(t\right)+\Omega_{q}^{2}\phi_{q}\left(t\right)+\Omega_{q}^{2}\int_{0}^{t}dt^{{}^{\prime }}M_{q}\left(t-t^{{}^{\prime }}\right)\partial_{t^{{}^{\prime }}}\phi_{q}\left(t^{{}^{\prime }}\right)=0$$
with the ideal memory kernel denoted as $M_{q}^{id}\left(t\right)$:(4)$$M_{q}^{id}\left(t\right)=\int d\vec{k}V\left(\vec{q};\vec{k},\vec{p}\right)\phi_{k}\left(t\right)\phi_{p}\left(t\right)$$
and
$$V\left(\vec{q};\vec{k},\vec{p}\right)=\rho S_{q}S_{k}S_{p}{\left[\left(\vec{q}\cdot \vec{k}\right)c_{k}+\left(\vec{q}\cdot \vec{p}\right)c_{p}\right]}^{2}/\left[2{\left(2\pi \right)}^{3}q^{4}\right]$$
with $\vec{p}=\vec{q}-\vec{k}$, $c_{q}=\left(1-1/S_{q}\right)/\rho$ and $\Omega_{q}^{2}=q^{2}k_{B}T/mS_{q}$. In such a way, the theory under some mean field approximation is able to predict the dynamic of the density correlator ϕ(t) as function of time *t*. These equations exhibit, at a critical temperature $T_{c}$, the bifurcation for $\phi_{q}\left(t\to \infty \right)=f_{q}$ also referred as the nonergodic transition [10]. For $T>T_{c}$, the correlator relaxes towards $f_{q}=0$ as expected in ergodic liquid states. Instead, density fluctuations for $T⩽T_{c}$ arrest in a disordered solid, quantified by a Debye–Waller factor $f_{q}>0$. This schematic MCT approach has opened the way to treat a variety of models, beyond the description of the standard glass transition with ϕ(t) that obeys the following equation:(5)$$\phi_{q}\left(t\right)+t_{0}\overset{\cdot }{\phi_{q}}\left(t\right)+\int_{0}^{t}M\left[\phi_{q}\left(t-s\right)\right]\overset{\cdot }{\phi_{q}}\left(s\right)ds=0$$
with $t_{0}$ as a microscopic time scale, $\overset{\cdot }{\phi_{q}}$ the $\phi_{q}\left(t\right)$ time derivative and the memory kernel *M* as a functional polynomial in ϕ(t). Taking the first three order terms, it is given by
$$M\left[\phi_{q}\left(t\right)\right]=v_{1}\phi_{q}\left(t\right)+v_{2}^{2}\phi_{q}\left(t\right)+v^{3\phi_{q}3}\left(t\right)$$
where $v_{1}$, $v_{2}$ and $v_{3}$ are coupling coefficients encoding the molecular interactions and are assumed to be smoothly increasing functions of external control parameters such as ρ, *P* or *T*. Hence,
(6)$$\phi_{q}\left(t\right)+t_{0}\phi_{q}\left(t\right)+v\int_{0}^{t}\phi_{q}\left(t-s\right)\phi_{q}\left(s\right)ds=0$$
where *v* as the control parameter. Depending just on $v^{\prime }s$, MCT defines different models with different properties [10,44]. More precisely, for q=1 it is defined the continuous model A ($v_{2}=v_{3}=0$), whereas for q=2 ($v_{1}=v_{3}=0$) we have the model B, characterized at the transition by an order parameter discontinuity. In that case the corresponding dynamic behavior is characterized by a two-step relaxation time typical of the glass transition.

It has been suggested that such model B is well reproduced by Fredrickson and Andersen facilitated Ising model [45] on a Bethe lattice (BL) [46], whose dynamics is well described in terms of a cluster approach based on the Bootstrap percolation model (BP) [47]. For $v_{3}=0$, we have the F12 model [10,44] and the phase diagram in the ($v_{1},v_{2}$) plane shows a line of continuous transition and a line of discontinuous transition, which join at a higher order critical point. Finally, for $v_{2}=0$, we have the F13 model, with $v_{1}$, $v_{3}$ as the control parameters [44].

As shown in Figure 3, the corresponding phase diagram ($v_{1},v_{3}$ plane) shows two transition lines. The first one is a straight horizontal line corresponding to a continuous glass transition, (like in model A). On crossing this line, by increasing $v_{1}$, f=ϕ(t→∞) increases continuously. The second line is a discontinuous transition, like in model B. This line (characterized by logarithm decay of the relaxation functions; see e.g., Figure 4) ends at the cusp-like A3 singularity [10], where the plateaus of the two transitions coincide. This situation is very important because it represents a molecular system made up of a repulsive hard core plus an attractive interaction [34].

It has also been conjectured that while the discontinuous line corresponds to a glass transition, and continuous line can be interpreted as a sol-gel transition [26]. A re-entrant glass transition due to the existence of two glassy states: one dominated by repulsion (with a caging structural arrest) and the other by attraction (with bonding arrest due to bonding) have been revealed by dynamic light scattering in polymethylmethacrylate (PMMA) depleted colloids dispersed in *cis*-decalin [36]. The short-range attraction was induced by adding a non-adsorbing polystyrene, so that the experiment was made at ambient temperature for different colloid volume fractions and polymer concentration. Although the measured self Intermediate Scattering Function (ISF; proportional to the coherent density correlator $\phi_{q}\left(t\right)$) appears compatible with a logarithmic decay, neither an attractive glass transition line nor the point A3 has been shown.

A theory datum point is the prediction of a critical temperature $T_{c}$ (or a critical concentration $C_{c}$) where the ergodic to nonergodic transition takes place, and $\phi_{q}\left(t\right)$ tends to a finite plateau $f_{q}$ for t→∞. The separation parameter is $\sigma \approx \left(T_{c}-T\right)/T_{c}$ (or $C_{c}-C)/C_{c}$), and are also proposes various density relaxation regimes with very different temporal scales. The short-time dynamics region $t<t_{0}$ is dominated by microscopic motions, and is followed by the β relaxation region that satisfies the scaling relation:(7)$$\phi_{q}\left(t\right)=f_{q}+h_{q}B\left(t\right)=f_{q}+c_{\sigma }h_{q}g_{\pm }\left(t/t_{\sigma }\right)$$
where $f_{q}$ is the so called non-ergodicity parameter (the plateau value), $h_{q}$ is the critical amplitude and B(t), that includes the time dependence, represents the β correlator. This latter equation holds for all correlators between variables which have an overlap with density fluctuations. B(t) depends singularly on two scales in the time and the control parameter: an amplitude or correlation scale $c_{\sigma }={\left|\sigma \right|}^{1/2}$ and the time scale $t_{\sigma }=t_{0}{\left|\sigma \right|}^{-1/2a}$. The initial part ($g_{\pm }\left(t\ll t_{\sigma }\right)=\left(t/t_{\sigma }\right)-a$), dominated by $t_{\sigma }$, gives the approach to $f_{q}$, while the final one follows the von Schweidler law $g_{-}\left(t_{\sigma }\ll t\ll t_{\sigma }^{\prime }\right)=-h_{q}{\left(t/t_{\sigma }^{\prime }\right)}^{b}$, with $t_{\sigma }=t_{0}{\left|\sigma \right|}^{\gamma }$ as the second characteristic timescale.
γ=(1/2a)+(1/2b).
where *a*
(0<a<0.5) and *b*
(0<b<1) are non-universal exponents determined solely by the so-called exponent parameter λ,
(8)$$\lambda =\Gamma^{2}\left(1-a\right)/\Gamma \left(1-2a\right)=\Gamma^{2}\left(1+b\right)/\Gamma \left(1+2b\right)$$
where Γ is the Euler gamma function and λ is in turn determined by the static structure factor $S_{q}$. Finally, for $t>t_{\lambda }$ there is the α relaxation regime. For hard spheres, a=0.301 and b=0.545 have been measured. In our previous photon correlation spectroscopy (PCS) experiment on a colloidal like system by considering the measured ISF, $F_{s}\left(q,t\right)$ we obtained b=0.6 and λ=0.7. In addition, a universal plot, for different ISFs, of the von Schweidler law gives γ=2.3 [35]. MCT (and the explanation of β relaxations) was developed originally by using cage effects (or clustering) that determines the temporal (or frequency) dependence of relaxations [10]. At very short times, density relaxation reflects the localized motion of individual particles, entailing the details of the hydrodynamic interactions. At longer times, particles are trapped in their neighbor structure.

Figure 5 illustrates the different characteristic MCT time regions. The corresponding ISFs were measured by PCS experiments made in a colloidal suspension of a PAMAM d5 NH${}_{2}$ dendrimer in methanol, at different φ. The von Schweidler universal plot obtained from these (Figure 5) is illustrated in Figure 6 (b=0.59 and γ=2.5).

As said, many aspects of the ideal MCT predictions have been detailed, experimentally or by means of MD simulations, in different systems. More recently, MCT showed also explanatory capabilities for colloidal glasses characterized not only by hard-core repulsion but also by attractive interactions, where new processes have been discovered and will be discussed in the next section.

It must be emphasized that, in the frame of the F13 model, of particular interest is a colloidal system characterized by an attractive contribution to the potential [34], and for the presence of a well-defined percolation line (whose properties had to be explicitly dependent on volume fraction φ and *T*). Therefore, much attention has been devoted, as we will see in detail below, to the water solution of polymer (Pluronic L64), in particular by the use of many different experimental techniques including rheology, neutron and light scattering [35,37,48,49].

Such a system is a combination of polyethylene oxide (PEO) and polypropylene oxide (PPO): (PEO)${}_{13}$(PPO)${}_{30}$(PEO)${}_{13}$. Being that both PEO and PPO are hydrophilic at low temperatures, L64 chains dissolve in water and exist as unimers. Instead, as *T* increases, the hydrogen-bond formation between water and polymer molecules decreases with PPO becoming rapidly less hydrophilic than PEO, and the copolymers acquire surfactant properties, aggregating in spherical micelles. Furthermore, at higher temperatures, water becomes a progressively poorer solvent for both PPO and PEO chains, giving rise to an inter-micellar attractive interaction (the system behaves like a grafted colloid). The evidence for the increased short-range micellar attraction as a function of *T* comes from the existence of a critical point at (C≈0.05 or φ≈0.098 and T=330.9 K) with a splinodal line and the sol-gel transition the percolation line.

Figure 7 illustrates the T−φ phase diagram for a D${}_{2}$O solution of such a system obtained by means of several techniques. This makes for a proper comparison between the neutron scattering data with those obtained with different techniques; hence, all the experiments on L64 here reported have D${}_{2}$O as solvent. In particular are shown the cloud point line (CP), the critical micellar concentration (CMC), the sol-gel transition line (PT), the kinetic glass transition boundary, the regions of the repulsive and attractive glass and finally the A${}_{3}$ locus. Figure 7 also shows details of the cusp-like singularity exhibiting glass–liquid–glass re-entrant behavior and the attractive-to-repulsive glass transition line beginning where the two branches cross and terminating at $A_{3}$ ($\varphi (A_{3})=0.544$), beyond which the long-time dynamics of the two glassy states become identical.

Theoretically [34,35,37], the phase behavior of a system like that is characterized by an effective temperature $T^{*}=k_{B}T/u$, the volume fraction of the micelles φ, and the fractional attractive well width Θ=Δ/d, where $k_{B}$ is the Boltzmann constant, −u is the depth of the attractive square well, *d* is the full particle diameter and Δ is the width of the well ($\Delta =d-d^{\prime }$, with $d^{\prime }$ as the hard core diameter). Hence, for a given Θ, aside from φ, as in the case of a pure hard sphere system, the effective temperature $T^{*}$ is introduced into the description of the phase behavior of the system as a second external control parameter, and thus the arrested glassy state can take place by increasing either φ or $T^{*}$.

Neutron scattering experiments have been used to evaluate $T^{*}$ as a function of the normal temperature *T* at different φ [37,48,49] showing that, as *T* increases, $T^{*}$ increases. By measuring the micellar aggregation number *N* as a function of *T* at different φ, it has been shown that the degree of self-association increases as *T* increases at a given φ. This is consistent with the fact that the PPO core becomes less hydrophilic at higher *T*. In addition, at a given temperature, *N* decreases as φ increases, indicating that *u* increases as φ increases. Finally, by comparing the effective temperatures obtained from fitting the experimental data in the liquid and glass states, it can be seen that the depth of the square well increases as *T* and φ increase, making a liquid–glass transition possible. Its phase diagram for the glass region is reported in Figure 8 in terms of $T^{*}$ and φ, and for different attractive well width.

As it can be observed, the full F13 condition is obtained for Θ=0.03; a continuity is also observable between the percolation line (or gel line) and the liquid attractive glass transition. The latter situation will be central in subsequent discussions because it represents the first indication of the possibilities of basic interrelationships between percolation models and MCT.

The next two figures propose how the singular behavior of this cusp-like singularity was measured by means of specific heat (Figure 9) and neutron experiments (Figure 10). In the first case, all the kinetic transitions of the L64/D${}_{2}$O system are detailed by means of a specific heat measurement, made through a differential calorimeter, in a volume fraction region (0.535<φ<0.55), just around the singularity A${}_{3}$. The narrow peaks at low *T* deal with the liquid glass transition. The peak observed near T=310 K is due to the glass-to-glass transition. In the top panel are detailed the following transitions: sol-gel, gel-attractive glass (AG), attractive (AG) to repulsive (RG) glass and, finally, the repulsive glass (AG) to gel. The central one represents the system evolution just near A${}_{3}$, from attractive to repulsive glass. The bottom panel instead shows a situation dominated only by repulsive glass.

Neutron scattering, small angle in particular (SANS), represents a powerful tool to trace the details of the liquid–glass transition line and the glass–glass transition. In the case of a monodisperse micellar (like the L64/D${}_{2}$O), the neutron scattering absolute intensity can be expressed by:$$I\left(q\right)=cN{\left(\sum_{i}b_{i}-\rho_{w}v_{p}\right)}^{2}P\left(q\right)S\left(q\right)$$
where *c* is the polymer number density, *N* is the aggregation number of polymers in a micelle, the sum refers to the coherent scattering lengths of atoms including a polymer molecule, $\rho_{w}$ is the scattering length density of D${}_{2}$O, $v_{p}$ is the polymer molecular volume, P(q) is the normalized intra-particle structure factor (giving the normalized form factor as $P\left(q\right)={\left|F\left(q\right)\right|}^{2}$) and S(q) is the inter-particle structure factor. Assuming the micelle to have a compact spherical hydrophobic core of a radius *a*, consisting of all the PPO segments with a dry core and a diffusive corona region consisting of PEO segments and solvent molecules, it is possible to apply an original model called the “cap and gown” [48] to calculate F(q).

If the mentioned square well potential (hard sphere with adhesive surface layer) is used, the inter-micellar attractive interaction can be evaluated. In such a way the absolute measured scattered SANS intensity can be fitted uniquely with four parameters: the aggregation number *N*, φ, Θ and $T^{*}$. In addition, whereas $\tilde{P}\left(q\right)$ is function of *N* only, the static structure factor S(q) depends on all four parameters. In these terms, a series of SANS experiments were carried out, in this copolymer micellar system, to examine the local structure of the glassy states, both in the region of the re-entrant glass transition and in the vicinity of the A${}_{3}$ point [37,48,49].

The SANS absolute intensity of an amorphous state, reflecting the local structure, is characterized by a single peak located at $q_{\max}$ (or a unique length scale $\Lambda =q_{\max}^{-1}$, which is the mean distance between mesoparticles). In addition, the SANS absolute I(q) distribution of a two-phase system (solvent and micellar aggregates) is proportional to a three-dimensional Fourier Transform of the Debye correlation function, which in the present case must be of the form Γ(r/Λ). Thus, by a simple transformation of variables, the dimensionless, scaled intensity distribution can be reduced to a unique function of a scaled wave vector in the form:$$\frac{q_{\max}^{3}I\left(Q\right)}{\langle \eta^{2}\rangle }=\int_{0}^{\infty }dx4\pi x^{2}j_{0}\left(xy\right)\Gamma \left(x\right)$$
where $x=q_{\max}r$, $y=q/q_{\max}$, and $\langle \eta^{2}\rangle$ is the so-called scattering invariant. In such a way by plotting, at a given φ, the scaled intensity as the scaled variable *y*, all the scattering intensity distribution at different temperatures within a single phase region should collapse into a single master curve. In this way, the distinct local structures associated with different phases occurring at different *T*-ranges can be identified. A scaling plot of the SANS intensities at φ=0.495 is proposed in Figure 10.

From the width of the peak it is possible to infer the system disorder degree. In particular, the data shown indicate, depending on *T*, two distinct degrees of disorder: whereas the narrower peak, which is resolution limited, represents the glassy state, the broader one (much broader than resolution) represents the liquid state characterized by a wider distribution of inter-particle distances respect to the glass. In such a way it has been detailed, by means of neutron scattering experiments, the re-entrant glass to glass transition and the cusp like A${}_{3}$ singularity [37,49]. In closing this part, it must be mentioned that all the scaled intensities are characterized by a unique length scale and collapse into one single master curve (*T* independent) showing the identical local structure of the two glasses. All of this constitute evident proof of the accuracy of MCT predictions and of its basic scientific validity.

### 2.2. Basic PT

The percolation model is a powerful tool of statistical physics in order to understand a large class of different phenomena. It has been used to describe a large variety of structures, such as composite materials, polymers and gels, proteins and biological properties, etc., and to analyze the corresponding dynamic properties. Similarly studied were the corresponding mechanical behaviors, conductivity, electrical breakdown and transport. Percolation represents an example of the scaling theories. Its definition is simple [29]: within a large lattice, each site of it can be randomly occupied with a probability *p*, or left with a probability p−1. A cluster is represented by a group of occupied neighboring sites. Every site of the cluster is connected with another of the same cluster, by at least a chain of sites. Occupied sites of different clusters are not connected. For small *p* there are in the system few large clusters and many isolated sites. The increase of *p* leads to largest aggregate that will comprise most of the occupied sites and touch all system sides. This happens for a unique critical threshold $p_{c}$, and the corresponding spanning cluster is named “*incipient infinite cluster*”. This is the bond percolation.

By defining as $n_{s}$ the ratio between the number of clusters containing σ sites and the number of sites in the whole lattice, *p* represents the fraction of sites of the largest cluster and the susceptibility as $\bar{\chi }=\sum_{s}\left(s^{2}n_{s}\right)$, which give a measure of the mean cluster size it is possible to represent, through a generalization oh the Fisher droplet model [50,51], the cluster numbers in terms of the critical phenomena scaling functions [52]:(9)$$n_{s}=s^{-\tau }f\left(\left(p-p_{c}\right)s^{-\sigma }\right)$$
which is an equation depending on two variables $x=s^{-1}$ and $y=p-p_{c}$, with $z=\left(p-p_{c}\right)s^{\sigma }$ as the scaled variable describing the behavior of the cluster numbers (thus *p* is the control parameter and $p_{c}$ its threshold value). Hence, $\bar{\chi }$ can be related to the critical exponents τ and σ. By replacing the sum by an integral, we have (above the threshold):$$\bar{\chi }=\int_{0}^{\infty }dss^{2}n_{s}=\int_{0}^{\infty }dss^{3-\tau }f\left(z\right)/s=\sigma^{-1}{\left(p-p_{c}\right)}^{-\gamma }\int_{0}^{\infty }dz^{\gamma -1}f\left(z\right)/s$$
with γ=(3−τ)/σ being the integral, the constant is:(10)$$\bar{\chi }∼\sigma^{-1}{\left(p-p_{c}\right)}^{-\gamma }$$

By considering that the probability of a site belongs to a cluster with *s* sites is $n_{s}$, it follows that the percolation probability *P* is zero below the threshold, and above it is:$$P=p-\sum_{s}\left(sn_{s}\right)$$
and thus
$$\begin{array}{ccc} \frac{dP}{dp} & = & 1-\sum_{s}sdn_{s}/dp=1-\sum_{s}s^{1-\tau }df_{z}/dp=1-\sum_{s}s^{\sigma +1-\tau }df_{z}/dz=\\ & = & 1-\int_{0}^{\infty }dss^{\sigma +1-\tau }df_{z}/dz=1-\sigma^{-1}{\left(p-p_{c}\right)}^{\beta -1}\int_{0}^{\infty }dzz^{(2-\tau )/\sigma }df_{z}/dz \end{array}$$
(11)whereβ−1=(2−τ−σ)/σorβ−1=(τ−2)/σ

Near the threshold the system has a correlation length ξ, that in terms of the gyration radius $R_{gs}$ of a cluster with *s* sites is:(12)$$\xi^{2}=\sum_{s}R_{gs}^{2}s^{2}n_{s}/\sum_{s}s^{2}n_{s}$$

Also at the threshold, ξ diverges as $\xi ∼{\left(p-p_{c}\right)}^{-v}$, where for a *d*-dimensional is dv=γ+2β. According to these exponents’ relations, knowing two of them all the others can be determined. That property of percolation clusters is typical of phase transitions and can be studied by assuming the temperature as control parameter ($y=T-T_{c}$). A proof of the validity of what is proposed is provided by viscosity data, of the L64/D_2_O colloidal suspensions characterized by a precise PT line, measured by a strain controlled rheometer by using a double wall cuette geometry, at different temperatures and volume fractions.

Figure 11 illustrates such a situation showing the corresponding viscosity data as a function of the reduced temperature $|T-T_{c}|$. The slope of the continuous straight line give v≃1.09 in according with the theoretical predictions for a 3-dimensional system [52].

All PT exponents are universal quantities depending only on the dimensionality of the embedding space [50,51]. The stochastic addition in a network of either vertices (sites) or connections (bond) leads to the observation of the PT, a structural change with the appearance of a connected component encompassing a finite fraction of the system. Percolation has always been regarded as a substrate-dependent but model-independent process, in the sense that the corresponding critical exponents are determined by the geometry of the system, and thus are identical for the bond and site percolation models.

## 3. Revised MCT Including Hopping Processes and Fragile to Strong
Dynamic Crossover (FSC)

A basic mathematical complexity prevents direct tests of the full MCT for the standard glass formers [53,54]. One way to apply MCT beyond its asymptotic scaling is in the so-called schematic models [31] that reduce the large set of coupled MCT equations to only a few (just one or two). By dropping the *q* dependence, introducing fit parameters and replacing the connection between S(q) and the theory’s coupling vertices, a quantitative agreement with experimental data is often obtained.

Schematic MCT models also provide a flexible testing ground for extensions of MCT as for example the inclusion of hopping processes. The β regime identification, explained by means of simple power laws has been the first major success of the ideal MCT of the glass transition, and the scaled relaxation was identified as the “*cage effect*” (where nearest-neighbor steric hindrance slows down the collective motion).

Taking into account the corresponding relaxations as a function of the frequency (rather than of the time), a minimum can be observed, at high *T*, whose position identifies the cage relaxation, and the α-peak as the relaxation mode by which particles escape the cages. MCT puts a theoretical basis to the power law ($\omega^{-b}$) describing the initial cage-escape dynamics, known long before as von Schweidler’s law [55]. These asymptotic scaling forms are demonstrated by various relaxation and scattering data around the β-minimum at not too low *T* [10]. In addition, one has distinguished glass formers that instead of the slow β-peak show an anomalous “wing” at this high-ω side of the α-peak, but there now appears to be a consensus that both cases should be attributed to the same phenomenon [56].

It was also noted that the standard ideal MCT description of glassy dynamics is no longer fully adequate for $T<T_{c}$, and not just for the fact that $T_{c}$ is typically found significantly above the phenomenological glass transition $T_{g}$. Meanwhile, for $T>T_{c}$, MCT successfully and quantitatively describes available data [10]. Below $T_{c}$, the theory captures only limited aspects of the dynamics, such as a nontrivial rise in the α-relaxation strength [57,58,59,60] or the boson peak occurring at high frequencies [61]. A possible divergence of $\tau_{\alpha }$ as $T\to T_{c}$ is not observed and consequently the model in such a standard form cannot be used to describe the complex α-relaxation shapes below $T_{c}$. For tracer diffusion studies in metallic melts [62], $T_{c}$ marks the system change from liquid-like transport ($T>T_{c}$) to a more complicate (hopping-mediated solid-like) transport below.

Below $T_{c}$, diffusion coefficients ($D_{s}\left(T\right)$) of tracer atoms differ greatly and essentially follow Arrhenius laws, whereas above (($T>T_{c}$), the values merge and follow, within experimental accuracy, the same super-Arrhenius *T*-dependence, typical of the collective caging motion. MCT extensions including additional “hopping” relaxation processes have been proposed soon [23], although they have not been applied. Empirical scaling laws have been used [63,64], as well as schematic models [65] that include an ad hoc hopping parameter. The used equations there have also been re-derived in order to reconcile MCT and the random first-order theory of the glass transition [66]. These studies were limited to discussing scaling deviations in the β-relaxation due to hopping. Recently, Chong [25] derived a semi-empirical microscopic MCT with hopping term, and was able to make parameter-free predictions that have been able to explain the decoupling of diffusion processes from the collective motion (the SEV). Moreover, features of the fragile-to-strong crossover found in a number of experiments have been analyzed within this theory [67,68].

Some extended-MCT models have been questioned [69], but this critique does not appear to apply strictly to the form used in the above-mentioned models [23,25]. On following, such a critical assessment an alternative hopping model has been proposed [70], whereas another approach has also been proposed based on a continued-fraction representation of the correlation function [71]. Together with this representation of the hopping-extended MCT [25,67], another formulation has also been proposed, by introducing a phenomenological time-dependent hopping rate, able to reproduce the experimental susceptibility spectra in the full frequency and temperature range available [72]. In the following, we consider both these hopping models.

### 3.1. Hopping in the Memory Kernel

Such an approach starts from the consideration that a fragile-to-strong dynamic crossover (FSC) phenomenon has been observed in confined water where the β-relaxation time [73] and the inverse of the self-diffusion [74] cross over from a non-Arrhenius to an Arrhenius behavior. On the other hand, it has also been recognized that the viscosity data on various glass-forming liquids exhibits an FSC-like feature at a crossover temperature $T_{x}$: while the *T* dependence of the viscosity of many glass-forming liquids can be well described by a power law for $T>T_{x}$, it smoothly crosses over to approximately an Arrhenius behavior for $T<T_{x}$ [75,76].

Recent measurements of diffusion constants in bulk glass-forming alloys display the crossover from a non-Arrhenius to an Arrhenius behavior [77,78]. This onset of a sharp increase in η(T) by the FSC has been also related to the MCT associating the temperature $T_{x}$ with the critical MCT temperature $T_{c}$). This situation is characteristic of a very large number of glass forming liquids where in particular the viscosities of these liquids collapse into a single one curve only by assuming $T_{x}$ as the only characteristic temperature [21]. Thus, the FSC seems a common, or at least an unexceptional, phenomenon in glass-forming systems, which contrasts the traditional view that a glass-forming liquid can be classified either as fragile or strong [79].

The account of these new experimental findings (in particular the FSC) is in the extended model [23,25]. By closely following the original MCT proposal [10] (see e.g., Equations (3)–(5), it is considered the correlator Laplace transform (LT) $\phi \left(z\right)=i\int_{0}^{\infty }dt\phi \left(z\right)\exp\left(izt\right)$:(13)$$\phi_{q}\left(z\right)=-{\left[z-\frac{\Omega_{q}^{2}}{z+\Omega_{q}^{2}M_{q}^{id}\left(z\right)}\right]}^{-1}$$
and the corresponding equations are the previous ZM (see e.g., Equations (4) and (5)), but with the memory kernel modified as:(14)$$M_{q}\left(z\right)=M_{q}^{id}\left(z\right)/\left[1-\delta_{q}\left(z\right)M_{q}^{id}\left(z\right)\right]$$
where the new contribution $\delta_{q}\left(z\right)$, called the “hopping kernel”, takes hopping into account. Such an extended kernel $M_{q}\left(z\right)$ deals with the interplay of two effects. Nonlinear interactions of density fluctuations, as described by the idealized $M_{q}^{id}\left(z\right)$), lead to the cage effect with a trend to produce arrested states for $T⩽T_{c}$. Instead, the hopping contribution $\delta_{q}\left(z\right)$, lead to the α relaxation and restore ergodicity at all temperatures.

In analogy with the ideal case, also the extended-MCT equations can be reformulated in terms of the longitudinal current correlator $K_{q}\left(z\right)$ in the form: $\phi_{q}\left(z\right)=-1/\left[z+K_{q}\left(z\right)\right]$, is $K_{q}^{id}\left(z\right)=\Omega_{q}^{2}/\left(z+\Omega_{q}^{2}M_{q}^{id}\left(z\right)\right)$. Combining the Laplace transform of the ZM equation (Equation (Equation 14)), with the new form of memory kernel, one finds [31]:(15)$$K_{q}\left(z\right)=\delta_{q}\left(z\right)-\frac{\Omega_{q}^{2}\left[1+R_{q}^{\left(2\right)}\left(z\right)\right]}{z+\left[1+R_{q}^{\left(1\right)}\left(z\right)\right]\Omega_{q}^{2}M_{q}^{id}\left(z\right)}$$
where $R_{q}^{\left(1\right)}\left(z\right)=-z\delta_{q}\left(z\right)/\Omega_{q}^{2}$ and $R_{q}^{\left(2\right)}\left(z\right)=z\delta_{q}\left(z\right)\left[1-\delta_{q}\left(z\right)M_{q}^{id}\left(z\right)\right]/\Omega_{q}^{2}$ are renormalization functions. Being that these latter functions are unimportant in the low-frequency regime z→0 (or long times) of interest, neglecting these terms, the function $K_{q}\left(z\right)$ obtains the form:$$K_{q}\left(z\right)=\delta_{q}\left(z\right)+K_{q}^{id}\left(z\right)=\delta_{q}\left(z\right)-\frac{\Omega_{q}^{2}}{\Omega_{q}^{2}M_{q}^{id}\left(z\right)}$$

By dropping $\delta_{q}\left(z\right)$, this equation reduces to that of the idealized MCT, $K_{q}\left(z\right)=$$K_{q}^{id}\left(z\right)$: approaching $T_{c}$ from above, $M_{q}^{id}\left(z\right)$ becomes larger due to the cage effect and the current correlator $K_{q}\left(z\right)$ vanishes at $T=T_{c}$, originating the sharp non-ergodicity. In the presence of $\delta_{q}\left(z\right)$, $K_{q}^{id}\left(z\right)$ becomes unimportant when $M_{q}^{id}\left(z\right)$ is large and there holds $K_{q}\left(z\right)\approx \delta_{q}\left(z\right)$.

The hopping kernel takes over and hinders the currents from vanishing, preventing the complete arrest of the density fluctuations. Hence, a dynamics crossover occurs at $T\approx T_{c}$ from the cage-effect-dominated regime ($K_{q}\left(z\right)=$$K_{q}^{id}\left(z\right)$) to the hopping one, $K_{q}\left(z\right)\approx \delta_{q}\left(z\right)$. A crossover implication for the α-relaxation time $\tau_{q}$ of $\phi_{q}\left(t\right)$, can be evaluated from $\tau_{q}∼\int_{0}^{\infty }dt\phi_{q}\left(t\right)$, i.e., $i\tau_{q}∼\phi_{q}\left(z\to 0\right)$. Further, $K_{q}\left(z\to 0\right)∼i/\tau_{q}$ is:(16)$$1/\tau_{q}\approx 1/\tau_{q}^{hop}+1/\tau_{q}^{id}$$
where $\tau_{q}^{hop}$ comes from the hopping via $\delta_{q}\left(z\to 0\right)=i/\tau_{q}^{hop}$. Thus, MCT predicts the dynamic crossover in the β-relaxation time $\tau_{q}$ at $T\approx T_{c}$ from $\tau_{q}\approx \tau_{q}^{id}$ to $\tau_{q}\approx \tau_{q}^{hop}$.

By marking the tagged particles, and related quantities, with the superscript or subscript ‘*s*’, we have that ZM equation for the tagged-particle density correlator is written as $\phi_{q}^{s}\left(t\right)=\langle \rho_{q}^{s*}\left(q,t\right)e^{iŁt}\rho_{q}^{s}\left(q,0\right)\rangle /\langle |\rho_{q}^{s}{|}^{2}\rangle$ and idealized-MCT memory will be $M_{q}^{sid}\left(t\right)=\int d\vec{k}V^{s}\left(\vec{q};\vec{k},\vec{p}\right)\phi_{k}^{s}\left(t\right)\phi_{p}^{s}\left(t\right)$ with $V^{s}=\rho S_{k}{\left[\left(\vec{q}\cdot \vec{p}\right)c_{k}\right]}^{2}/\left[{\left(2\pi \right)}^{3}q^{4}\right]$ [80]. Moreover, all the related functions can be given in these terms. By considering that: (i) the self-diffusion coefficient $D_{s}$ in terms of the velocity correlation ($K^{s}\left(t\right)=\langle {\vec{v}}_{s}\left(t\right){\vec{v}}_{s}\left(0\right)\rangle /3$) is $D_{s}=\int_{0}^{\infty }dtK^{s}\left(t\right)$; (ii) and $K^{s}\left(t\right)$ is related with the mean-squared displacement ($\delta r_{s}^{2}\left(t\right)=\langle {\left[{\vec{r}}_{s}\left(t\right)-{\vec{r}}_{s}\left(t\right)\right]}^{2}\rangle$) as $K^{s}\left(t\right)=\left(1/6\right)\partial_{t}^{2}\delta r_{s}^{2}\left(t\right)$. The extended-MCT equations for $K^{s}\left(t\right)$ can be deduced from those for $\delta r_{s}^{2}\left(t\right)$, by exploiting the small-*q* behavior of $\phi_{q}^{s}\left(t\right)=1-q^{2}\delta r_{s}^{2}\left(t\right)/6+O\left(q^{4}\right)$ [79], with a resulting ZM representation:$$\partial_{t}K^{s}\left(t\right)+v^{2}\int_{0}^{t}dt^{{}^{\prime }}M^{s}\left(t-t^{{}^{\prime }}\right)K^{s}\left(t^{{}^{\prime }}\right)=0$$
with $v^{2}=k_{B}T/m$, and $M^{s}\left(z\right)=M^{sid}/\left(1-\delta^{s}\left(z\right)M^{sid}\right)$; here $M^{s}\left(z\right)=lim_{q\to 0}M_{q}^{s}\left(t\right)$ and $\delta^{s}\left(z\right)=lim_{q\to 0}\delta_{q}^{s}\left(z\right)/q^{2}$. Thus:$$K^{s}\left(z\right)=\delta^{s}\left(z\right)-\frac{v^{2}\left[1+R^{s(2)}\left(z\right)\right]}{z+\left[1+R^{s(1)}\left(z\right)\right]v^{2}M^{sid}\left(z\right)}$$

Also in this case $R^{s(2)}\left(z\right)$ and $R^{s(1)}\left(z\right)$ are unimportant in the low-frequency regime z→0 ($R^{s(2)}\left(z\right)=-z\delta^{s}\left(z\right)/v^{2}$; $R^{s(1)}\left(z\right)=z\delta^{s}\left(z\right)\left[1-\delta^{s}\left(z\right)M^{sid}\left(z\right)\right]/v^{2}$ ). Neglecting these terms, it is $K^{s}\left(z\right)=\delta^{s}\left(z\right)+K^{sid}\left(z\right)$, with $K^{sid}\left(z\right)=-v^{2}/\left(z+v^{2}M^{sid}\left(z\right)\right)$. Being $K^{s}\left(z\to 0\right)=iD$:(17)$$D_{s}\approx D_{s}^{id}+D_{s}^{hop}$$

Thus, the extended MCT predicts the dynamic crossover in the self-diffusion constant $D_{s}$ at $T\approx T_{c}$ from D≈$D_{s}^{id}$ to $D_{s}^{hop}$.

Figure 12 (top panel) reports the density correlators calculated at the peak position (q=7.3) of the static structure factor S(q) for different *T*, in the case of a Lennard–Jones system (with S(q) obtained within the Percus–Yevic approximation) by means of this extended MCT (solid curves) together with their ideal counterparts (dashed curves). The ideal correlators are characterized by the non-ergodicity plateau (and the appearance of the critical point); instead, in the extended case, the plateau disappears in favor of a long time decay due to the hopping processes. Illustrated at the bottom is the self-diffusion *D* versus $T_{c}/T$. Also here, the solid curves deal with the extended MCT, and dashed curve refers to $D^{id}$ from the ideal MCT (which vanishes at $T_{c}/T=1$ according to the predicted power law $D^{id}\approx {\left|T-T_{c}\right|}^{\gamma }$ [10]. The dash-dotted curve represents $D^{hop}$ due to the hopping processes.

### 3.2. Hopping in the Correlator

In this case the correlator LT (Equation (Equation 13)) has been amended with a new time-dependent hopping rate [72] so that the corresponding equation of motion can be written as:(18)$$\phi \left(z\right)=-{\left[z+\Delta \left(z\right)-\frac{\Omega^{2}}{z+i\omega +\Omega^{2}M_{q}^{id}\left(z\right)}\right]}^{-1}$$
again a form derived from the ZM projection-operator formalism [42], M(z) is the memory kernel of fluctuating forces (the core target of MCT’s approximations by reflecting that for short-times Ω and ω can be considered constants). Consequently:(19)$$K_{q}\left(z\right)=\Delta \left(z\right)-\frac{\Omega^{2}}{z+i\omega +\Omega^{2}M_{q}^{id}\left(z\right)}$$
The difference between these two models lies in the fact that, while in the first case the memory kernel is amended (Equation (Equation 14)), in this second one it is just the correlator (Equation (Equation 18)); by setting Δ(z)=0 and approximating the memory kernel M(z) as a nonlinear functional of the correlator in the time domain, M(t)=F[ϕ(t)], the original schematic MCT is obtained.

In this frame the model able to reproduce the asymptotic power laws predicted by the microscopic theory is the F12 model, specified by $v_{3}=0$ ($M\left(t\right)=v_{1}\phi \left(t\right)+v_{2}\phi^{2}\left(t\right)$). The quadratic term is essential: it gives rise to bifurcations in the long-time limits of the correlators, $f=lim_{t\to \infty }\phi \left(t\right)$, (the idealized-MCT glass transition). In these conditions the coupling coefficients change smoothly, at critical values ($v_{1}^{c},v_{2}^{c}$), but *f* jumps discontinuously from zero to a finite value $f_{c}$ (the nonergodicity of the idealized glass). The exponent parameter $\lambda \in \left[1/2,1\right]$ determines the non-universal power-law exponents *a* and *b* governing the long-time dynamics close to the transition.

In the F12 model, one has $v_{1}^{c}=\left(2\lambda -1\right)/\lambda^{2}$, and $v_{2}^{c}=1/\lambda^{2}$. The transition gives rise to the ϕ(t) two-step decay, with an initial $\phi \left(t\right)=f_{c}+h{\left(t/t_{0}\right)}^{-a}$ toward the asymptotic plateau $f_{c}$, and then the von Schweidler decay $\phi \left(t\right)=f_{c}+h^{\prime }{\left(t/t_{\sigma }\right)}^{-b}$ (from the plateau toward zero on the liquid side of the transition). Here, $t_{0}$ is the time scale of the short-time motion, whereas $t_{\sigma }$ represents a diverging time scale upon approaching the transition. In the ω domain, these two asymptotes translate to the low- and high-frequency power laws around a β-minimum. Hence, the idealized MCT transition at ($v_{1}^{c},v_{2}^{c}$)(liquid side) is accompanied by a second divergent relaxation time in ϕ(t), the α-relaxation time that governs the correlation functions long-time decay.

The central point of schematic extended-MCT models is to incorporate additional relaxation processes (not captured by the M(t)) that restore ergodicity inside the glass state. In the idealized MCT, Δ(z) describes an additional slow relaxation channel (a very small correction due a not completely arrested cage effect). If the primary MCT relaxation channel vanishes, any contribution to Δ(z), also small, will dominate the long times and thus must be accounted for. By transforming ϕ(z), Equation (Equation 18), in the time domain, it is:(20)$$\overset{\cdot \cdot }{\phi }\left(t\right)+C_{\Delta \overset{\cdot }{\phi }}\left(t\right)+\left[\Delta \left(t\right)+\overset{\cdot }{\phi_{0}}\left(t\right)\delta \left(t\right)\right]+\omega \overset{\cdot }{\phi }\left(t\right)+vC_{\Delta \phi }\left(t\right)+\Omega^{2}\phi \left(t\right)+\Omega^{2}C_{M\overset{\cdot }{\phi }}\left(t\right)+C_{\Delta M\phi }\left(t\right)=0$$
$C_{AB}\left(t\right)$ represents the convolution of two (or more) functions A(t) and B(t) like e.g., f(t) and g(t)$C_{fg}\left(t\right)=\int_{0}^{t}f\left(t-\tau \right)g\left(\tau \right)d\tau$, whereas $\overset{\cdot }{\phi }\left(t\right)$ and $\overset{\cdot \cdot }{\phi }\left(t\right)$ are the correlator temporal derivatives. Equation (Equation 19) can be solved under the initial conditions: ϕ(0)=1, $\partial_{t}{\phi \left(t\right)|}_{t\to 0}=-\Delta \left(0\right)=\partial_{t}\phi_{0}\left(t\right)$; by approximating $\Delta \left(t\right)\approx \delta_{0}\delta \left(t\right)$ as $\Delta \left(z\right)\approx i\delta_{0}$, leads to the extended-MCT model, also in the context of *q*-dependent correlators.

It must be noted that the correspond long times (low frequencies) behaviors are not affected by such a change. To describe the non-exponentiality of the relaxation below $T_{c}$ the Δ time dependences are relevant. For $t\gg t_{\sigma }$, the cage-effect channel has already attained its long-t limit, ϕ(z)=−1/(z/f+Δ(z)) and in the time domain under the initial condition ϕ(t)=f it is:(21)$$\partial_{t}\phi \left(t\right)=-f\int_{0}^{t}\Delta \left(t-t^{{}^{\prime }}\right)\phi \left(t^{{}^{\prime }}\right)dt^{{}^{\prime }}$$
which is an approximation that neglects the Δ(z) in the memory kernel and thus does not capture the exact long-t behavior. However, any empirical distribution of relaxation times, such a stretched exponential, can be reported in that expression if the generalization is admitted [81]
(22)$$\Delta \left(t-t^{{}^{\prime }}\right)\approx \Delta \left(t,t^{{}^{\prime }}\right)\approx \hat{\Delta }\left(t^{{}^{\prime }}\right)\delta \left(t-t^{{}^{\prime }}\right)$$

Hence, $\phi \left(t\right)∼f\exp\left[-R(t)\right]$, where $R\left(t\right)=\int_{0}^{t}f\hat{\Delta }\left(t^{{}^{\prime }}\right)dt^{{}^{\prime }}$. $\hat{\Delta }$ is the so called hopping contribution, a form able to capture the physics of dynamic heterogeneities below $T_{c}$. The idea to describe the density fluctuations relaxation as a random process has been suggested to be central in MCT [82]; this approximation leads to a significant simplification:(23)$$\overset{\cdot \cdot }{\phi }\left(t\right)+\left[\omega +\hat{\Delta }\left(t\right)\right]\overset{\cdot }{\phi }\left(t\right)+\left[\Omega^{2}+\omega \hat{\Delta }\right]\phi \left(t\right)+\Omega^{2}C_{M\overset{\cdot }{\phi }}\left(t\right)+\Omega^{2}\hat{\Delta }C_{M\phi }\left(t\right)\left(t\right)=0$$

A correlation functions equation formally derived by using the modified projection operators introduced by Mori and Tokuyama (MT) [83]. In a single-correlator model, the $\hat{\Delta }\left(t\right)$ time dependence is responsible for the stretching of the hopping-induced relaxation, and its overall magnitude controls the corresponding time scale. The previous kernel model, by assuming localized hopping events embedded in the glassy structure [25], also derives an Arrhenius rate where the strength of the frozen matrix—its elastic modulus—enters the effective barrier height for the activated events.

In this formalism of the linear response functions, the longitudinal elastic modulus (E) exhibits a thermodynamic contribution that varies smoothly across the MCT transition, and a non-ergodic contribution essentially given by the MCT memory kernel M(t→∞)=M[f], according to Ref. [84], that arises in the glass only, meaning that:(24)$$\tilde{f}\hat{\Delta }\left(t\right)\approx \phi^{2}\left(t\right)\exp\left(-EM\left[\tilde{f}\right]/\rho k_{B}T\right)$$
where $\tilde{f}=f$ in the glass, whereas $\tilde{f}=f_{c}$ in the liquid. This situation highlights the way with which MCT also accounts for the system viscoelasticity. In dense hard-core repulsive liquids [84] a typical value is $E/\rho k_{B}T\approx 50$. Smaller E values will enhance the hopping contribution and M[f] increases as one enters deeper into the glass. Precise form of the Arrhenius factor can be derived by estimating these hopping rates [85]. The quadratic dependence of $\hat{\Delta }\left(t\right)$ on ϕ(t) could be replaced by a linear one, more appropriate for intermittent hopping relaxations [70]. This would result in less stretching of the final relaxation. It seems that the correct form cannot be decided within a schematic model. However, numerical findings for the schematic model (with memory kernel and hopping kernel) qualitatively explains recent findings for hard-sphere-like colloids [86].

The kernel model also pays attention to the probe-density fluctuations and specific Kohlrausch relaxation rates. In the first case (as described by the extended F12-model correlator), the corresponding coupling is simply expressed by writing a second correlator $\phi_{A}^{s}\left(z\right)$ for the fluctuation of a dynamic variable *A* probed in the experiment [87]. This could be the reorientation of permanent dipoles, the molecular rotational and diffusional motions, or describe the *q*-dependent coupling of incoherent scattering to the collective density fluctuations in neutron scattering. The LT of $\phi_{A}^{s}\left(z\right)$ determines the corresponding dynamic susceptibility $\chi_{A}$:(25)$$\chi_{A}\left(z\right)/\chi_{A}=z\phi_{A}^{s}\left(z\right)+1$$

It is interesting to focus on the normalized susceptibility spectra, $\chi_{A}^{\prime \prime }\left(\omega \right)/\chi_{A}$ where $\chi_{A}$ can be treated as a *T* independent adjustable parameter, so that the memory kernel $M_{A}^{s}\left(z\right)$ is:(26)$$M_{A}^{s}\left(z\right)=v_{A}^{s}\phi \left(t\right)\phi_{A}^{s}\left(t\right)$$

The subscript A can deal with the analysis of dynamic light scattering data, (A = LS), or dielectric spectroscopy data, (A = DS). The hopping motion will be present both for the collective density ϕ(t), and the probe $\phi_{A}^{s}\left(t\right)$ correlator. In the case of the combined α- and slow β-relaxations, they are better modeled by means of both terms, as:(27)$${\tilde{f}}^{s}{\hat{\Delta }}^{s}\left(t\right)\approx \phi \left(t\right)\phi_{A}^{s}\left(t\right)\exp\left(-EM\left[\tilde{f}\right]/\rho k_{B}T\right)$$
by assuming that the *t*-dependence is given by probe-density correlations, and the escape rate (even of the single particle) is still due to collective rate. ϕ(t) embodies the fact that hopping relaxation in glass formers is a highly collective effect, as seen (above and below $T_{c}$) in the tracer diffusion of metallic glass formers [88]. Both dynamic susceptibilities $\chi^{\prime \prime }(\omega )$ from the correlators ϕ(t) and $\phi_{A}^{s}\left(t\right)$ have been calculated.

The corresponding ϕ(t) comparison with a Debye-relaxation (${\chi^{\prime \prime }}_{Debye}\left(\omega \right)\propto \omega \tau /\left[1+{\left(\omega \tau \right)}^{2}\right]$) reveals the features of non-exponential relaxation even inside the ideal glass state. Comparing with the results including hopping, one finds that the stretching almost stays constant as one crosses the glass transition [24]. The susceptibility spectra follow an analogous trend. The $\chi_{A}^{\prime \prime }\left(\omega \right)$ stretching is less, and the α-relaxation peak strength is enhanced for the choice of large coupling strength. In the glass region, there is the splitting of the relaxation peak into two contributions. A result due to the coupling between $\phi_{A}^{s}\left(t\right)$ and ϕ(t).

The relaxing ϕ(t) induces relaxation in $\phi_{A}^{s}\left(t\right)$, but on the same time scale the hopping term (slightly slower) constitutes a second mode giving rise at two time scales in the $\phi_{A}^{s}\left(t\right)$ decay: the shorter one corresponds to the collective fluctuations decay, and induces a shoulder in the high-ω side of the α-relaxation peak in $\chi_{A}^{\prime \prime }\left(\omega \right)$. This shoulder bears some reminiscence of a slow β-peak that emerges as one moves deeper into the idealized-glass state.

The schematic model with ad hoc t-dependent hopping rates represents an example of how to generate stretched decay inside the MCT glass state, and how density and probe correlators contribute to the slow β-peak in the presence of two distinct rate kernels. There are, however, some drawbacks in the model respect the experimental data: the spectra broadening (in particular in the β-peak (or HF-wing) region) is much less than that observed. This is due to the hopping rate form, $\hat{\Delta }\left(t\right)\approx \phi {\left(t\right)}^{2}$ and the appearance of a deep minimum in the spectra for $T<T_{c}$ that is not consistent with experimental data [89].

The slow β-modes interpretation as the interplay between collective relaxation and probe-variable relaxation is still consistent within the extended MCT, providing that:(28)$${\hat{\Delta }}_{KWW}\left(t\right)=\beta \frac{t^{\beta -1}}{f\tau^{\beta }}$$

Such a long time stretched-exponential relaxation (KWW) [90] is empirical and the original MCT is in some way compatible with such a correlation functions limiting form [80]. In addition, ${\hat{\Delta }}_{KWW}\left(t\right)$ diverges as t→0, since the stretched-exponential relaxation has no regular short-time expansion. To make the model well defined, it has been assumed in Equation (Equation 19) that:(29)$$\hat{\Delta }\left(t\right)={\hat{\Delta }}_{KWW}\left(1/\Omega \right),\;\;\mathrm{for}\;\;\;t<1/\Omega \;\;\;\mathrm{and}\;\;\;\;{\hat{\Delta }}_{KWW}\left(t\right),\;\;\mathrm{for}\;\;\;\;t\geq 1/\Omega$$
This is a reasonable approach, as the model is only intended to be applied for times larger than the time scale of microscopic motion set by 1/Ω. A corresponding expression can be also used for the probe correlator.

The model introduces as fit parameters (Ω,v) and ($\Omega_{A}^{s},v_{A}^{s}$), used to model the microscopic dynamics and to fix the frequency and energy units. The low-frequency part of the solutions has a negligible dependence on their, up to a rescaling of frequencies. Thus, $v_{1},v_{2}$ become the crucial control parameters driving the collective correlator ϕ(t) through its idealized glass transition. $v_{A}^{s}$ determines the strength for the coupling between probe and collective motion [87]. Finally, the hopping parameters, (τ,β) and ($\tau_{A}^{s}$,$\beta_{A}^{s}$) that determine the shape of the α-relaxation below $T_{c}$, are crucial in fitting this part of the data.

Figure 13 is an example for the interplay between ϕ(t) and $\phi_{A}^{s}\left(t\right)$. The fit of experimental data (covering up to 18 orders of magnitude) is a demanding test of any model. In the actual case, the use of the two correlators (ϕ(t) and $\phi_{A}^{s}\left(t\right)$) with the interplay between these two hopping processes is unique to the MCT ansatz (Equation (Equation 23)). The possibility to consistently fit two different data sets, A=LS or A=DS, using the same parameter set for the underlying ϕ(t), is a nontrivial reality. An example are the obtained findings in the fit of the experimental spectra [91], far below $T_{c}$ showed in the following [72].

## 4. The Interplay between Percolation and MCT

In the context of the present work MCT and PT have been, separately, the subject of many studies. Both practically constitute the main tool for understanding the chemical physics of complex systems in terms of scale laws. While PT is at the root of the problems concerning the sol-gel transition, MCT has certainly clarified many of the complexities of dynamic arrest. Only recently, however, has it been attempted to understand common mechanisms [26,30,47,92,93,94].

By changing the control parameters, many physical systems reach a slow dynamics regime followed by an arrested or a quasi-arrested state; examples, among others, are gels and glasses, both considered amorphous solids. Gels are elastic disordered solids observed at low density in systems, whereas molecules are bonded to each other through attractive forces or chemical links.

The percolation (sol-gel) transition is a continuous transition. By using the random percolation an explicit scaling form for the dynamic correlator was found, and general scaling laws connecting the dynamic exponents can be derived [26,30]. It has been proven that using mean field percolation exponents the same scaling form for the correlator and scaling relations were valid for the continuous transition of MCT model A [10]. Such a result suggests that the origin of the continuous MCT power laws is due to an underlying static transition in the same universality class of the random PT.

In this frame it was considered, as paradigmatic example, the Fredrickson and Andersen facilitated Ising model [45] on a BL [46]; with ϕ(t), that tends to the order parameter of the bootstrap percolation (BP) model [95]. The BP model exhibits a mixed order transition with an order parameter which discontinuous jumps at the transition. Nevertheless, the fluctuations, and the associated critical length, diverge as the transition is approached from the glassy phase.

Generalizing the cluster approach, the dynamic behavior for the correlator (and the susceptibility) of the FA facilitated model was predicted [26] (by including universal scaling laws that relate dynamic exponents with the universal BP exponents). This proposes that, by means of the mean field values of these static exponents, the dynamic behavior and the scaling laws are the same of the MCT model B [26,96,97]. On considering the results found for the continuous transition by using clustering concept, a new, more precise form is found for the approach of the correlator to the plateau, characterized by a power law and followed by a stretched exponential divided by a power law.

Predictions were numerically verified on both FA and MCT model B, suggesting a general common mechanism for discontinuous glass transition at mean field level, based on a static transition in the same universality class of BP with a diverging static length, responsible for the origin of scaling and universality present in such a wide range of systems, apparently different from each other. In the frame of the classical two step relaxation (β and α), the correlator can be written as
(30)$$\phi_{q}\left(t\right)≃f_{q}+\varepsilon^{\beta }F\left(t/t_{\beta },t/t_{\alpha }\right)$$
where β=1/2 is the order parameter BP exponent, $t_{\beta }∼\varepsilon^{-z_{1}}$ corresponds to the first step relaxation to the plateau, and $t_{\alpha }∼\varepsilon^{-z}$ corresponds to the second relaxation time. At the criticality it is $\phi_{q}\left(t\right)-f_{q}∼t^{-a}$, with $z_{1}=\beta /a=1/2a$.

The plateau approach is given by a stretched exponential divided by a power law and the departure from it is $\phi_{q}\left(t\right)-f_{q}∼-\varepsilon^{\beta }{\left(t/t_{\beta }\right)}^{b}$. This latter situation is interpreted as a propagating damage from an initial density of altered sites $\varepsilon^{\beta }$, times ${\left(t/\tau_{\beta }\right)}^{b}$ (the number of distinct damaged sites by one initial altered site during the time *t*).

The scaling function of the two variables have as consequence the scaling relation between dynamic exponents a,b,z and the BP static exponent β, z=β/a+β/b=1/2a+1/2b. The dynamic susceptibility, $\chi_{4}\left(t\right)=N\left(\langle q^{2}\left(t\right)\rangle -\langle q(t)\rangle \right)$ (where *N* is the number of particles), is in the liquid phase $\chi_{4}\left(t\right)=\varepsilon^{-\gamma }G_{-}\left(t/t_{\beta },t/t_{\alpha }\right)$, where γ=1 is the BP critical exponent of the order parameter fluctuations. This scaling leads to $\chi_{4}\left(t\right)\;∼t^{a\gamma /\beta }=t^{2a}$ for t<τβ with a crossover to $t^{2b}$ for $t_{\beta }<t\ll t_{\alpha }$.

This is a consequence that the dynamics in this regime receive due the damage to propagation and that $\chi_{4}\left(t\right)$ is proportional to the square of distinct damaged sites. Finally, $\chi_{4}\left(t=\tau_{\beta }\right)\;∼\varepsilon^{-\gamma }=\varepsilon^{-1}$ and $\chi_{4}\left(t=\tau_{\alpha }\right)\;∼\varepsilon^{-\gamma -2\beta }=\varepsilon^{-2}$ and goes to zero in the infinite time limit. In the glassy phase, $\chi_{4}\left(t\right)\;∼t^{a\gamma /\beta }=t^{2a}$ for $t<\tau_{\beta }$ with a crossover to a constant plateau diverging as $\varepsilon^{-\gamma }=\varepsilon^{-1}$.

### 4.1. Kinetic Facilitated Models and Bootstrap Percolation

The system dynamic is characterized by the correlator, ϕ(t)=〈q(t)〉. So that the related susceptibility, $\chi_{4}\left(t\right)=N\left(\langle q^{2}\left(t\right)\rangle -{\langle q(t)\rangle }^{2}\right)$ and pair correlation function, $g_{ij}\left(t\right)$, are:(31)$$q\left(t\right)=\frac{1}{N}\sum_{i}n_{i}\left(t\right),\;g_{ij}\left(t\right)=\langle n_{i}\left(t\right)n_{j}\left(t\right)\rangle -\langle n_{i}\left(t\right)\rangle \langle n_{j}\left(t\right)\rangle ,\;\chi_{4}\left(t\right)=\frac{1}{N}\sum_{ij}g_{ij}\left(t\right)$$
with $n_{i}\left(t\right)=0,1$ depending whether a spin at site *i* has flipped or not in the interval (0,t) [94].

On a BL of coordination number z=k+1, the model, for 0<f<k−1, has a transition from a high-*T* liquid (whit a large density of down spins), to a frozen phase at low *T*, (whit few down spins and the onset of an infinite cluster of blocked spins) [98]. In that t→∞ limit [46], this transition corresponds exactly to that of BP. Hence, BP has a mixed order transition (its percolation order parameter *P* jumps discontinuously at the threshold from zero to $P_{c}$), the fluctuation χ of the order parameter and the associated length ξ diverge as:(32)$$P-P_{c}∼\varepsilon^{\beta },\;\chi ∼\varepsilon^{\gamma },\;\gamma ∼\varepsilon^{v}$$
β=1/2,γ=1,v=1/4
Thus, for the FA model in the glassy phase, m=P, $\chi_{4}\left(\infty \right)=\chi$, and $\xi_{4}\left(\infty \right)=\xi$. With $m=lim_{t\to \infty }\phi \left(t\right)$ and $\chi_{4}\left(\infty \right)=lim_{t\to \infty }N\left(\langle q^{2}\left(t\right)\rangle -{\langle q(t)\rangle }^{2}\right)$ as the FA order parameter and its fluctuation, respectively, and $\xi_{4}\left(\infty \right)$ the associated length.

#### 4.1.1. Decay to the Plateau by Using the Cluster Approach

Coming from the glassy phase $T<T_{c}$, the FA the static properties exhibits a mixed order transition at $T_{c}$ whose critical behavior given by the Equation (Equation 33) scaling laws. By re-defining the order parameter as $f-f_{c}$, the transition can be considered as a continuous one and a cluster formalism can be applied, such as the sol-gel transition and the dynamic transition of the MCT model A [30,47,92]. In this approach, the system can be described by a distribution of clusters n(s), where each cluster of size *s* decays exponentially $\phi_{s}\left(t\right)∼e^{-t/\tau_{s}}$, with $\tau_{s}$ as the relaxation time of a cluster of size *s*. The larger the cluster size, the larger is $\tau_{s}$; thus, it is natural to assume (as for polymer systems) the power law:(33)$$\tau_{s}∼s^{x}$$
with *x* as a constant. Being that the entire system density correlator is the sum over all clusters
(34)$$\phi_{s}\left(t\right)-f_{q}≃\sum_{s}sn\left(s\right)e^{-t/\tau_{s}}$$
$n\left(s\right)∼s^{-\tau }e^{-s/s^{*}}$ is the cluster distribution associated to the BP fluctuation; τ=2+β/(β+γ) and $s^{*}=\epsilon^{-(\beta +\gamma )}$, and finally β=1/2 and γ=1 are the mean field BP exponents. In the sol-gel transition and in the MCT model A, the cluster distribution is thus given by the random percolation with β=1 and γ=1 [29].

Such a general approach is based under the only assumption that the system configuration can be partitioned in a clusters distribution, each decaying with a relaxation time proportional to $s^{x}$. This approach is still valid, just like in a liquid–gas transition close the critical point, and able to describe the system criticality in terms of a droplet distribution, such as Fisher’s droplet model [50,51]. In the glassy phase, $T\leq T_{c}$, the cluster formalism of the continuous transition can be applied, a formalism which also predicts a pure power law decay [30] for the entire range of times at $T=T_{c}$, and the same power law below $T_{c}$, provided that $t\ll \tau_{\beta }$,
(35)$$\phi \left(t\right)-f_{q}∼t^{-a},\;t\ll \tau_{\beta },\;\tau_{\beta }∼\varepsilon^{-z_{1}}$$
$$a=\frac{1}{x}\frac{\beta }{\beta +\gamma },\;z_{1}=x\left(\beta +\gamma \right)=\frac{\beta }{a}$$
being that *x* is related to the relaxation time of a size *s* fluctuation, and with the BP exponents:$$a=\frac{1}{3x},\;z_{1}=\frac{3x}{2}=\frac{1}{2a}$$

As in the continuous case, close $T_{c}$ the power law is followed by a transient, given by a stretched exponential combined with a power law [30]:(36)$$\phi \left(t\right)-f_{q}∼\varepsilon^{\beta }{\left(\frac{\tau_{\beta }}{t}\right)}^{c}e^{-{\left(t/\tau_{\beta }\right)}^{\gamma }}$$
$$c=\frac{3\beta +\gamma }{2(x+1)(\beta +\gamma )}=\frac{5}{6(x+1)}=\frac{5a}{2(1+3a)}$$
and
$$\begin{array}{ccc} y & = & \frac{1}{x+1}=\frac{3a}{1+3a}\\ \frac{c}{y} & = & \frac{3\beta +\gamma }{2(\beta +\gamma )}=\frac{5}{6} \end{array}$$

Large scale numerical simulations of the FA model on the lattice (BL) with k=3, f=2 and N=2, fully agree with these cluster approach predictions. a≈0.29 was obtained from the power law decay at the $T_{c}$ and from this value, the exponents y≈0.46 and c≈0.39, have been obtained. In the liquid phase, there is the same approach to the plateau.

#### 4.1.2. Departure from the Plateau Using the Damage Spreading Mechanism

In the liquid phase, the clusters (when the plateau is still present) survive on time scales of the order of $\tau_{\beta }$. The small clusters start to decay first and the last relaxing clusters are the largest ones, i.e., the critical clusters. In such a situation the relaxed sites, act as initial altered (damaged) to “free” the sites of the potential boot-strap cluster represented by the plateau. As *t* increases, this damage spreads through a cascade process [99], so that the potential infinite cluster (which contributes to the plateau) is a sea of quasi frozen sites, surrounded by critical clusters.

Above $T_{c}$ these latters eventually decay, whereas just below $T_{c}$, they become frozen and part of the infinite cluster. The number of sites m(t) in the core is related to the correlator as $\phi \left(t\right)≃f_{c}-m\left(t\right)$, where $m\left(t\right)∼\varepsilon^{\beta }{\left(t/\tau_{\beta }\right)}^{b}$, β=1/2, and $\varepsilon^{\beta }$ represents the density of sites in the critical clusters (the density of initial damaged sites). $1/\tau_{\beta }$ is the diffusion coefficient of the sites in the critical cluster, being *b* a dynamic exponent related to the spreading damage mechanism. Finally, in the α regime,
(37)$$\phi \left(t\right)∼g\left(\frac{t}{\tau_{\alpha }}\right),\;\tau_{\alpha }∼\varepsilon^{-z}$$
and, like in MCT, using the matching conditions with the previous regime:(38)$$z=\frac{\beta }{a}+\frac{\beta }{b}=\frac{1}{2a}+\frac{1}{2b}$$
having the liquid phase two relaxations ($\tau_{\beta }$ and $\tau_{\alpha }$) the density correlator ϕ(t) must be:(39)$$\phi \left(t\right)-f_{q}≃\varepsilon^{\beta }F\left(\frac{t}{\tau_{\beta }},\frac{t}{\tau_{\alpha }}\right)$$
where $F\left(x,y\right)=F_{1}\left(x\right)$ for y≪1, $F_{1}\left(x\right)=x^{-\beta z_{1}}$ for x≪1, $F_{1}\left(x\right)=-x^{b}$ for x>1, and $F\left(x,y\right)=-x^{b}F_{2}\left(y\right)$ for x≫1 and y>1. The requirement that ϕ(t) for $t>\tau_{\alpha }$ is a function of $t/\tau_{\alpha }$ only, implying that $\varepsilon_{\beta }x^{b}=y^{b}$, which in turn leads to $\tau_{\alpha }=\tau^{b}\tau_{\beta }$ where $\tau_{b}∼\varepsilon^{-z_{b}}$ with $z_{b}=1/2b$. Taking into account that $\tau_{\beta }∼\varepsilon^{-z_{1}}$, it follows the scaling relation Equation (Equation 39).

### 4.2. Comparison with Discontinuous MCT Model B

The MCT model B correlator satisfies these scaling forms, suggesting a connection between such a PT picture and MCT. This is also validated on considering that the mean field static BP exponents coincide with those found in the Random Field Ising (RFI) model in an external field [39]. Proposed as well in the previous analysis, MCT predicts a relation between the exponent *a* and *b* and the MCT parameter λ. Instead, in the FA instead of λ we have *x* as parameter, so that:$$a=\frac{1}{x}\frac{\beta }{\beta +\gamma }=\frac{1}{3x}$$
and if the FA approach applies to MCT:$$\lambda =\Gamma^{2}\left(1-1/3x\right)/\Gamma \left(1-2/3x\right)=\Gamma^{2}\left(1+b\right)/\Gamma \left(1+2b\right)$$

MD data show that this MCT relation is well verified on the FA model, thus strongly supporting the idea that the FA model in mean field reproduces entirely MCT [96]. In the next subsection, we will refer to the FA model, but the same predictions apply to the MCT as well.

#### 4.2.1. Fluctuations of the Order Parameter

Dynamic heterogeneities play a basic role in understanding the glass transition [28,100,101,102,103]. They are described through the susceptibility, $\chi_{4}\left(t\right)$, defined as the fluctuations of the dynamic order parameter. As made for the correlator, we express $\chi_{4}\left(t\right)$ as a scaling function of two variables; since in the glassy phase for t→∞ it coincides with the fluctuation of the BP order parameter, diverging as γ:(40)$$\chi_{4}\left(t\right)≃\varepsilon^{-\gamma }G_{\pm }\left(\frac{t}{\tau_{\beta }},\frac{t}{\tau_{\alpha }}\right)$$
using (+) in the glassy and (−) liquid phase, respectively.

**Glass Phase**. In this phase $\tau_{\alpha }=\infty$ is:(41)$$\chi_{4}\left(t\right)≃\varepsilon^{-\gamma }G_{+}\left(\frac{t}{\tau_{\beta }},0\right)≃\varepsilon^{-\gamma }F_{+}\left(\frac{t}{\tau_{\beta }}\right)$$
being $F_{+}=const$ for t=∞, and for $t<\tau_{\beta }$ is $F_{+}\left(t/\tau_{\beta }\right)={\left(t/\tau_{\beta }\right)}^{a\gamma /\beta }$ with $\tau_{\beta }∼\varepsilon^{-\beta /a}$ is:(42)$$\chi_{4}\left(t\right)≃t^{a\gamma /\beta }=t^{2a}\;\mathrm{for}\;t<\tau_{\beta }\;\;\;\mathrm{and}\;\mathrm{thus}\;\varepsilon \;\mathrm{independent}$$
(43)$$\chi_{4}\left(t\right)∼\varepsilon^{-\gamma }=\varepsilon^{-1}\;\mathrm{for}\;t\to \infty$$

Hence, $\chi_{4}\left(t\right)$ grows as $t^{2a}$ up to the plateau ($t∼\tau_{\beta }$), whose value diverges as γ=1.

**Liquid Phase**. Here, for $t\ll \tau_{\alpha }$ is:(44)$$\chi_{4}\left(t\right)≃\varepsilon^{-\gamma }G_{-}\left(\frac{t}{\tau_{\beta }},0\right)≃\varepsilon^{-\gamma }F_{-}\left(\frac{t}{\tau_{\beta }}\right)$$

In the early regime, $t\leq \tau_{\beta }$, the behavior is the same as in the glassy phase,
(45)$$\chi_{4}\left(t\right)≃t^{a\gamma /\beta }=t^{2a}\;\mathrm{for}\;t<\tau_{\beta }\;$$
(46)$$\chi_{4}\left(t\right)∼\varepsilon^{-\gamma }=\varepsilon^{-1}\;\mathrm{for}\;t=\tau_{\beta }$$

In the late β regime, ($t_{\beta }<t\ll t_{\alpha }$) being the dynamic due to the damage spreading mechanism, $\chi_{4}\left(t\right)$ must be proportional to the square of the number of visited sites $m^{2}\left(t\right)∼t^{2b}$ (similarly to what is found in the diffusing defects mechanism [93]), therefore:(47)$$F_{-}\left(\frac{t}{\tau_{\beta }}\right)={\left(\frac{t}{\tau_{\beta }}\right)}^{2b}\;\;\mathrm{for}\;t_{\beta }<t\ll t_{\alpha }$$
(48)$$\chi_{4}\left(t\right)≃\varepsilon^{-\gamma -2}{\left(\frac{t}{\tau_{\beta }}\right)}^{2b}\;\mathrm{for}\;t_{\beta }<t\ll t_{\alpha }$$
where $\tau_{\alpha }∼\varepsilon^{-z}$, $\tau_{\beta }∼\varepsilon^{-\beta /a}$. In the late β and α regime $t>\tau_{\beta }$, is $G_{-}\left(t/\tau_{\beta },t/\tau_{\alpha }\right)={\left(t/\tau_{\beta }\right)}^{2b}H_{-}\left(t/\tau_{\alpha }\right)$, where $H_{-}\left(y\right)=const$, for y≤1 in order to match the behavior in the late β regime Equation (Equation 45), and goes to zero for y≫1, as $\chi_{4}\left(t\right)$ in the infinite time limit tends to the value of the BP susceptibility χ, which is zero in the liquid phase. Therefore:(49)$$\chi_{4}\left(t\right)=\varepsilon^{-\gamma -2\beta }{\left(\frac{t}{\tau_{\alpha }}\right)}^{2b}H_{-}\left(t/\tau_{\alpha }\right)\;\mathrm{for}\;t>\tau_{\beta }\;$$
and
(50)$$\chi_{4}\left(t\right)∼\varepsilon^{-\gamma -2\beta }∼\varepsilon^{-2}\;\mathrm{for}\;t=\tau_{\beta }\;$$
where β=1/2 and γ=1. The scaling relation 3.13 has been numerically verified (FA model on the BL), showing that in the β regime ($t\ll \tau_{\beta }$) all curves rescale onto a unique function corresponding to $F_{-}\left(x\right)$ (with x=$t/\tau_{\beta }$), and also by fully confirming Equation (Equation 45) [47]: with $\chi_{4}\left(t\right)∼t^{2a}$ in the early β regime and ∼$t^{2b}$ in the late one.

Figure 14 reports the data collapse of the rescaled susceptibility $\chi_{4}\left(t\right)$, showing that in the β regime $t\ll t_{\alpha }$ all curves rescale onto a unique function corresponding to $F_{-}\left(x\right)$ (with $x=t/\tau_{\beta }$, and that $\chi_{4}\left(t\right)$$∼\varepsilon^{-\gamma }$ for $t=\tau_{\beta }$ (with γ=1). In particular, the scaling relation γ=1, $\tau_{\beta }∼\varepsilon^{-1/2a}$ are shown. Straight lines show the power law behaviors in the early $t^{2a}$ and late β regime ($t^{2b}$), with a=0.29 and b=0.50.

#### 4.2.2. Comparison with MCT

$\chi_{4}\left(t\right)$ was also studied within the p-spin model and within the MCT theory using a diagrammatic approach [104]. The MCT results [93] predicted for $\chi_{4}\left(t\right)$ the growths $t^{a}$ and $t^{b}$ for the early and late β regime and a growth of the $t^{*}$ maximum with an exponent 1. Later it was argued [105] that this behavior is valid only for ensembles where all conserved degrees of freedom are fixed, e.g., Newtonian or Brownian dynamics. In the NVT ensemble, otherwise other diagrams would contribute to $\chi_{4}\left(t\right)$ leading to $t^{2a}$ and $t^{2b}$ and an exponent 2 for the growth of the $\chi_{4}\left(t\right)$ maximum of $t^{*}$.

In such a way, it has been substantially proposed that a cluster approach and the damage spreading, applied to the FA in the mean field BP, predict a discontinuous dynamic transition with the same scaling behavior found in the MCT discontinuous transition. A transition, this latter, characterized by a static mixed order transition (in the same universality class of the BP) with critical fluctuations diverging only in the glassy phase.

Nevertheless, the liquid phase dynamics is strongly influenced by this transition, as shown by the behavior of the $\chi_{4}\left(t\right)$ heterogeneities. Just this transition, characterized by a diverging static length, is responsible for the corresponding scaling laws and universality class. In this scenario the sol-gel transition (with a static random PT) is described by the continuous MCT model A in a different universality class. In conclusion, it can be stressed that an MD study conducted by using a DLVO potential give a further demonstration that both the glass and the sol-gel transitions can be described by an explicit MCT model [106] (the F13 MCT schematic model).

### 4.3. The Related Main Findings

#### 4.3.1. From Theory and MD Simulations

The common mechanisms between MCT and PT are essentially based on the universality present in both the F12 and F13 models [26,30,47,92,93,94]. As said, MCT and PT have been intensively studied by using both MD simulation and experimental approaches; here, however, we will refer to those specific results of the interrelation between MCT and PT, and above all to data both related to the FSC and viscoelasticity.

Research findings essentially on two types of systems, physical and chemical *gels*, are considered. The physical ones are colloids, essentially hard spheres (mono- [107] or poly-dispersed [108,109,110]) or charged (interacting via a DLVO potential, shielded or not [111,112]). On the other hand, the chemical gels belong to the large class of polymeric solutions where the gel phase is the state characterized by percolating cluster, and the gelation transition as the percolation line [5,7].

An MD study proposes that polymer suspensions can represent the F13 case [113]. By changing their volume fraction φ, the PT line (decreasing with φ) was found to occur at a bond probability $p_{gel}\left(\varphi \right)$, and the particle diffusivity evaluated for each value of the bonding probability *p*, (by assuming the glass line as $\varphi_{g}\left(p\right)$) follows the power law $|\varphi -\varphi_{g}{|}^{\gamma }$. However, the obtained phase diagram similar to that of the MCT F13 model does not show the A3 singularity. It is observed that the gel line (associated to the continuous transition) and the glassy line (to the discontinuous one) cross each other, with the glassy line entering in the gel region) and the liquid–glass transition becoming a gel–glass transition.

The sol-gel transition appears as a divergence in the system relaxation time τ(q) (obtained from the corresponding ISF, $F_{s}\left(q,t\right)$ as $\int dtF_{s}\left(q,t\right)t/\int dtF_{s}\left(q,t\right)$) at the smallest q. Regarding size, the structural arrest is comparable to the box size: at the threshold the spanning cluster is unable to diffuse, opposite of the finite ones [114].

In the glass state, instead of the particles being unable to leave the cage formed by neighbor particles, the structural arrest occurs on all length-scales. Important characteristics of such a situation along the glass line are [92]: (i) the logarithmic decays inside the gel phase; (ii) the self ISF on increasing *p* that shows a decreasing in the plateau value as an effect of the cage motion.

As predicted in the F13 model, the plateau value associated to the gel transition increases and the logarithm decay appears when the two plateaus coincide. The A3 singularity, typical of models where the two interfering arrested lines are both discontinuous glass transition lines [34], has been well studied experimentally, by using different techniques (including viscoelasticity), in attractive colloidal particles. Experimental work where the logarithmic decays were also found [35,37,48,49].

Physical gels, in comparison with the chemical one, having bonds with a finite lifetime, were extensively studied by MD simulations. An interesting case is a recent study [106] on systems with a DLVO interaction (between two particles *i* and *j*) represented as:(51)$$V_{ij}\left(0\right)=\epsilon \left[A{\left(\frac{\sigma_{ij}}{r}\right)}^{36}+B{\left(\frac{\sigma_{ij}}{r}\right)}^{6}+C\left(\frac{e^{-r/\xi }}{r/\xi }\right)\right]$$
where the temperature *T* is represented in ($\epsilon /k_{B}$) units, σ is the particle diameter, $\sigma_{ij}=\left(\sigma_{i}+\sigma_{j}\right)/2$, the wave vectors are in units of $\sigma^{-1}$ and the time is $\sqrt{m/\sigma^{2}/\epsilon }$. A=3.56, B=7.67, C=75.08, ξ=0.49, and $\sigma_{i}=\sigma_{j}$ ensure the monodispersity.

It must be noted that under monodispersed conditions, like in real colloids, transitions from this disordered cluster phase, to an ordered hexagonal lattice first, and to an ordered lamellar phase then, are observed by increasing φ. In the polydispersed case a disordered cluster phase is observed at low *T* and φ and on increasing the φ, just the polydispersity avoids the transition to the order, so that the system enters a “supercooled” metastable liquid phase until structural arrest (gel) occurs very close to the percolation threshold.

Moreover, the DLVO having an attractive tail, depending of the relative strength, can originate a clustering with finite lifetimes or a stable aggregation (coagulation). In this study, the cluster relaxation was detailed by means of the time autocorrelation function of bonds, B(t), evaluated as $n_{ij}\left(t\right)=1$ if particles *i* and *j* are linked at time *t*, otherwise $n_{ij}\left(t\right)=0$.

Hence, the corresponding $F_{s}\left(q,t\left(q\right)\right)$, the structural relaxation time, τ(q) and the bond lifetime $\tau_{b}$ are thus obtained from such a quantity. The resulting data, at low φ and small *q*, where the bond lifetime is larger than the structural relaxation time τ(q), obey to a power law, a regime for which the system seems made of permanent clusters like in chemical gels.

By increasing φ, when these lifetimes become comparable to each other, a new regime is observed, where, besides the clusters, also the particles crowding starts to play a role in the dynamic slowing down. Consequently, the relaxation time departs from the lower φ power law behavior and for φ>0.12, logarithmic decays appear in the self ISF, although the A3 singularity was not observed. In these results it is clearly proposed an important reality: the glass transition and the sol-gel transitions can be described by a particular MCT model.

The glass transition is characterized by a discontinuity of the order parameter (model B) and the sol-gel transition instead of by a continuity (model A). As such, a geometrical interpretation leads to associate the BP to the discontinuous transition of model B and the random percolation to the continuous transition of model A. Therefore, the same general formalism can treat the transition from a liquid to an arrested state, whether glass or gel.

The phenomenology observed in some polymer systems and associated to the interplay between chemical gels (with permanent bonds) and glasses, can be interpreted in terms of the F13 MCT model, clarifying also the origin of logarithmic decays observed in these systems. Giving the suggestion that there are other systems where such theory could be applied, e.g., colloidal gels, where bonds between particles have finite lifetime and the logarithmic decays appear.

It would be interesting to study such systems at higher concentrations in order to check if the MCT scenario applies also in DLVO colloidal gels. Moreover, glasses with pinning impurities deserve to be studied with these generalized MCT models; indeed, adding pinning impurities bootstrap percolation will crossover towards random percolation.

Finally, let us mention possible applications to water polymers mixtures (polyelectrolytes and biopolymers). Because polymers (or proteins) form bonds, once the PT threshold of bonded polymers are reached, the water system stops flowing, becoming an amorphous solid. As it is well known in the deep supercooled regime, water becomes a glass. It would be interesting to explore the interplay between the glass and the gel lines in these solutions, in particular in those of biological interest where extremely interesting phenomena occur.

Summarizing, by changing the control parameters, many systems reach a slow dynamics regime followed by an arrested or a quasi-arrested state. Examples, among others, are gels and glasses. Both are considered amorphous solids in the sense that can reach a slow dynamics regime followed by an arrested disordered state. Gels are elastic disordered solids observed at low density in systems, where molecules are bonded to each other through attractive forces or chemical links.

In chemical gels, the sol-gel transition was explained in terms of the appearance of a percolating cluster of monomers linked by bonds [5,7], and experimental measurements confirmed this interpretation. Glasses instead exhibit a structural arrest usually at high density, and glass transition is also observed in liquids interacting only with excluded volume.

The glass transition is considered a mixed order transition, with discontinuous order parameter, and MCT well describes dynamic behavior of glassy systems at least at mean field level [10,31]. As mentioned, the ideal description of glassy dynamics is no longer fully adequate for $T<T_{c}$, with $T_{c}$ typically found well above the glass transition $T_{g}$. A reasonable way to study the system dynamics below $T_{c}$ is that such it marks the transport change from liquid-like to an hopping-mediated solid-like, a situation suggested by the experimental evidence of the dynamic crossover at temperatures higher than the glass transition.

#### 4.3.2. From Experimental Data (a) Hopping in the Memory Kernel

Moreover, at given φ, decreasing *T*, the FSC is found. This is interpreted [38,48,115] as a crossover from MCT regime, where power law is expected, to the extended MCT one, where a hopping regime characterized by an exponential like behavior is expected. At the same time, due to the attractive force, the first regime may be attributed to the aggregation of percolation clusters, like in the sol-gel transition [21].

Interestingly, in some polymer or colloidal systems (e.g., PL64/D_2_O AHS micellar system [48]), a crossover from a gel-like to a glass-like behavior is observable by changing the control parameters. Such a system is also characterized by a viscoelastic behavior at low φ, where the viscosity is well fitted by a power law, and by increasing φ, a crossover to a different regime is observed (at the FSC). On these bases was considered the idea for an interplay between gel and glass [116] to describe the sol-gel transition using MCT, and simultaneously, to use percolation concepts in order to understand the criticality underlying the dynamic singularity found in MCT. In this frame, the F13 schematic model, where the continuous transition meets the discontinuous one, can be of relevant interest.

As said, the first extension of this schematic model was proposed by Chong only considering the relevance for the arrest process of the FSC observed in the PL64/D_2_O system [67,68], which consists of a semi-empirical microscopic MCT with hopping term, able not only to make parameter-free predictions but also to explains the underlying SEV. For this, of interest are the numerical results of a Lennard = −Jones (LJ) system whose particles interact via the potential $V\left(r\right)=4\epsilon_{LJ}\left\{\left(\sigma_{LJ}/r\right)12-\left(\sigma_{LJ}/r\right)6\right\}.$ With the static structure factor $S_{q}$, required for determining $\Omega_{q}^{2}$, $M_{q}^{id}$ and $M_{q}^{sid}$, evaluated within the Percus–Yevick approximation [117]. The corresponding $T_{c}$, was found to be $T_{c}=1.637$ for the average number density ρ=1.093 [25].

After these results, a model for the extended MCT was considered one for which the hopping processes are incorporated via a dynamic theory formulated to describe diffusion–jump processes in crystals [118]. In such a case hopping, with the rate $w_{hop}$, vibrational fluctuations occur in the quasi-arrested state with particles trapped inside the cages. Hence, both the collective and tagged-particle hopping kernels are then given by: $\delta_{q}\left(z\right)=$$\delta_{q}^{s}\left(z\right)/S_{q}$ and $\delta_{q}^{s}\left(z\right)=w_{hop}N_{c}\left[1-\sin(qa)/qa\right]/f_{q}^{s}$, with $N_{c}$ being the coordination number of the first shell surrounding a particle and *a* as the hopping distance.

The $\phi_{q}\left(t\right)$, was thus calculated (at the peak position $q^{*}=7.3$ of $S_{q}$) for both the idealized- and extended-MCT for some temperatures near $T_{c}$. The ideal $\phi_{q}^{id}\left(t\right)$ exhibits the bifurcation of the long-time limit at $T=T_{c}$ ($\phi_{q}^{id}\left(t\to \infty \right)=0$ for $T>T_{c}$) and for $T\leq T_{c}$ it is $\phi_{q}^{id}\left(t\to \infty \right)=f_{q}>0$. On the other hand, the hopping contribution $\phi_{q}^{hop}\left(t\right)$ relaxes to zero even for $T\leq T_{c}$. Moreover, the hopping processes start to affect density fluctuations above $T_{c}$. The α-relaxation time $\tau_{q^{*}}$, if evaluated under the same conditions, is considered as a $T_{c}/T$ function, reveals that $\tau_{q^{*}}^{id}$ diverges at $T_{c}/T=1$ according to the idealized theory bifurcation and such a divergence obeys to the power law $\tau_{q^{*}}^{id}∼{\left|T-T_{c}\right|}^{-\gamma }$.

In the extended MCT case, this law is replaced by a curve characterized by a smooth crossover very near $\tau_{q^{*}}^{id}$. A relevant characteristic of this α-relaxation time representation is that for $T>T_{c}$ it can be well fitted by super-Arrhenius power law, whereas in the opposite case of low temperatures $T\leq T_{c}$ the corresponding $\tau_{q^{*}}^{hop}$ is nearly Arrhenius. Thus, in the present model the α-relaxation time exhibits the fragile to strong dynamic crossover (FSC) at $T\approx T_{c}$, by confirming the singular energetic configuration represented by the “energy landscape” and the corresponding thermal evolution. It is relevant that these time values cover in the *T*-range of interest many orders of magnitude (about 6).

The related self-diffusion $D_{s}$ full confirms these findings. Both the $D_{s}^{id}$ and $D_{s}^{hop}$, show as a function of $T_{c}/T$, that the $D_{s}^{id}$ of the idealized MCT vanishes at $T_{c}/T=1$ as $D_{s}^{id}$$∼|T-T_{c}{|}^{\gamma }$ reflecting the predicted dynamic arrest. Instead, depending on the *T*-regime, $D_{s}$ is determined by the larger one of $D_{s}^{id}$ and $D_{s}^{hop}$, explaining why $D_{s}$ crosses over from $D_{s}\approx D_{s}^{id}$ to $D_{s}^{hop}$ at $T_{c}/T\approx 1$. In addition, that $D_{s}$ exhibits the FSC at $T\approx T_{c}$.

All of this being $D_{s}∼{\left(\tau_{q}^{*}\right)}^{-1}$ is fully reflected in the system SER. Obviously is $\tau_{q}^{*}∼\eta /T$. The SER is known to be accurate for normal- and high-temperature liquids, and holds also within the idealized MCT since it predicts that both $\tau_{q}^{*}$ and $1/D_{s}$ exhibit a universal power-law behavior $|T-T_{c}{|}^{-\gamma }$ for $T\to T_{c}+$ [10].

As already demonstrated, the extended MCT also predicts the SER breakdown near and below $T_{c}$ in agreement with experimental [21,119,120,121] and theoretical [122] observations in glass-forming liquids. As proposed in the log–log plots of Figure 15 in terms of $D_{s}$ versus $\tau_{q}^{*}$ (calculated by the extended MCT), can be observed that, while the SE holds at high *T* (e.g., dotted curve), it breaks near and below $T_{c}$.

Such a SE violation is fitted by a fractional relation $D_{s}∼{\left(\tau_{q}^{*}\right)}^{-x}$ with 0<x<1. Theoretical investigation of the SE violation based on kinetically constrained models [122], suggests the corresponding *x* value. The main result of is that, while in the fragile case is $D_{s}∼{\left(\tau_{q}^{*}\right)}^{-0.73}$ (weakly dependent on the dimensionality *d*), in the strong case the violation is sensitive to *d* ($D_{s}∼{\left(\tau_{q}^{*}\right)}^{-2/3}$ for d=1 and $D_{s}$$∼{\left(\tau_{q}^{*}\right)}^{-0.95}$ for d=3). Such an observation agrees with the experimental result for water confined in cylindrical tubes (d=1) for which the fractional relations with the exponents 0.73 and 2/3 are found to fit the data for the fragile and strong sides, respectively.

#### 4.3.3. From Experimental Data (b) Hopping in the Correlator

Results equally interesting were also obtained in terms of the hopping in the correlator. More precisely, the propylene carbonate (PPC) data coming from the experimental susceptibility spectra measured by dielectric loss spectroscopy (DS) and by depolarized light scattering (DL) at different *T* and in a very wide ω range (about 18 order of magnitude) were satisfactory fitted. In the related study a special focus was set on the $v_{1}$ and $v_{2}$ variation in the ($v_{1}$,$v_{2}$) space, in a representation that allows us to check the linear dependence $\gamma ∼(T_{c}-T)$, of DS and LS independent data by proving accurately the consistency of the model fits.

Very interesting are, for both the experiments, the *T*-behavior of the coupling strengths entering the memory kernel $v_{A}^{s}$ (Equation (Equation 26)). Both parameters are smoothly increasing functions on decreasing *T*. In coherence for the different rotational contributions probed by the two techniques, the collective density dynamics coupling for the DS probe is stronger than that versus LS. $v_{A}^{s}$ determine, below $T_{c}$, the appearance of the excess wing relative to the α-peak, meaning a consistent wing description in both data sets.

Moreover, below $T_{c}$, as confirmed by independent fits the hopping parameters τ and β entering ϕ(t) there is a good data agree. For the broadness of the wing excess (or slow β-peak) β decreases with decreasing *T*, resulting very small around $T_{g}$. Additionally, according to the t−T superposition principle, the stretching parameters $\beta_{A}^{s}$ entering the probe correlators are weakly T—dependent; for LS, its values, due to the correlation between the α-peak strength and stretching, are lower.

The hopping time cannot be modeled by simple Arrhenius behavior, this is consistent with Chong et al.’s results [67,68]. In that framework the plateau height enters the expression for the hopping rates, in particular the activation energy. This height at $T>T_{c}$ is T-independent, below it follows the square-root dependence, with an additional *T* dependence of the activation energy, leading to a crossover from a rather weak *T* dependence of the hopping rates to a high activation energy of the hopping times below $T_{c}$ [68].

At high *T*, the α-peak separates from the microscopic band, revealing an intermediate minimum in the GHz–MHz range; and the low- and high-ω sides of this minimum can be described by power laws $\omega^{-b}$ and $\omega^{a}$, respectively. Being that the main relaxation modes are driven by pair fluctuations, the caging can become so strong that, below $T_{c}$, no longer allow a relaxation to an ergodic liquid state. This implies $\tau_{\alpha }\to \infty$ and a white noise in the spectrum for low ω (χ(ω)∝ω). The idealized $T_{c}$ can be thus identified by the scaling forms predicted for the β-minimum position (and height), or that of the α-peak for $T>T_{c}$.

Following these suggestions a more empirical version of the model’s equations was considered to test specific ligth-scattering data of a glass former (PPC [123]). Specifically, by combining literature data of dielectric susceptibilities and depolarized Brillouin spectra (DLS) with those of photon correlation spectroscopy (PCS). A consistent description of all data sets is thus obtained by adjusting only few physical parameters. The wing excess (or slow β-relaxation commonly observed in the susceptibility spectra) was modeled as the effect of a coupling of the individual probe correlator to the collective density fluctuations.

Experimental susceptibility spectra of the glass former PPC (($T_{g}=157$ K, $T_{m}=218$ K)) are explored with a two-component MCT schematic model from above the melting point down to temperatures far below the $T>T_{c}$ [31].

By introducing a phenomenological time-dependent hopping rate, the spectra were reproduced in the full ω and *T* range available [24].

In PCS, customary measured is the autocorrelation function of the scattered intensity $g_{2}\left(t\right),$ related to the electric field function $g_{1}\left(t\right)$. In this system both the measured $g_{1}\left(t\right)$ and $g_{1}^{2}\left(t\right)$ have the long-time part of the decay well reproduced by a stretched exponential function, $g\left(t\right)\approx aexp[-{\left(t/\tau \right)}^{\beta }]$. These time PCS data can be combined with the frequency data (obtained with a Tandem Fabry–Perot setup [124]), if numerically Fourier-transformed:$$\chi_{LS}\left(\omega \right)=\frac{1}{2\pi }\int_{-\infty }^{+\infty }dtg_{1}\left(t\right)e^{-i\omega t}$$

Hence, the susceptibility data obtained from the PCS spectra can be compared with the cited frequency data after a properly normalization.

The transformed PPC spectra point out that some additional relaxation modes likely exist on the high-ω side of the measured α-peak. From the DS measurements, such contributions are the slow β-mode (or the wing excess). A contribution, this latter, that is rather small, in agreement with related findings obtained for salol, 2-picoline, and dimethyl phtalate [125,126]. Presented in Figure 16 are the proposed susceptibility spectra for PPC as measured by DS (left) and by DLS (right) at the indicated temperatures. Solid lines indicate fits with the extended schematic-MCT model, while the dashed ones are fits with the ideal MCT model.

As can be seen, the used model provides a consistent data fit giving an accurate account of all available data spanning up to 18 orders of magnitude in ω, and 3 orders of magnitude in amplitude, for temperatures ranging from around the melting temperature, down to below $T_{g}$. Other findings on this subject can derive from the normalized density-fluctuation correlation functions F(q,t); in fact, their corresponding susceptibility spectra χ(ω) can be used to detail the evolution of the supercooled liquid versus its glass phase in terms of the MCT models.

The susceptibility spectra χ(ω) evaluated from the F(q,t) are characterized, as well as χ(ω) spectra measured with direct techniques, by peaks with maxima and minima reflecting the properties of the primary α-relaxation that follow the MCT predictions. With decreasing *T*, their peaks are shifted to lower ω and a distinct minimum appears in the χ(ω) spectra. Moreover, for $T\to T_{c}$, the minimum position ($\omega_{min}$) is shifted to lower ω.

However, at $T_{c}$, the α−peak and the minimum do not disappear (as predicted by the ideal MCT) for the effect of hopping processes.

In addition, in this case, as far as for the viscosity (or the other transport parameters), the critical $T_{c}$ can be deduced from the *T*-dependence of $\chi_{min}$ (the χ value at $\omega_{min}$) by means of MCT forms: $\chi_{min}∼\varepsilon^{1/2}$ and $\omega_{min}∼\varepsilon^{1/(2a)}$. The near coincidence between $T_{c}$ and the dynamic FSC temperature $T_{L}$, evidenced for these molecular liquids by means of transport functions, can be also observed by an Arrhenius plot of the measured $\chi_{min}$.

Figure 16 shows this for DS spectra as function of the frequency, while Figure 17 proposes a similar situation obtained by means of the F(q,t) in the time regime with χ(q,t)=−dF(q,t)/dT. This is in particular for the ISF previously proposed and calculated in the case of a Lennard–Jones system (Figure 12 top panel). In such a case, the peak height of χ(q,t), denoted as $\chi^{*}\left(q,t\right)$ (and related to the size of the system dynamic heterogeneities [27,28]), increases as one approaches $T_{c}$, but below $T_{c}$ it decreases. The latter situation is attributed to the existence of the dynamic FSC and thus to related behaviors [121].

## 5. Viscoelasticity

### 5.1. MCT

The Equation (Equation 27) shows that MCT can account for the viscoelastic (and rheological) properties of a material. Thus, in this frame MCT has been intensively applied in many different areas of material science, including granular liquids or biology. As demonstrated in polymer science the system viscoelasticity is well characterized by the dynamic complex viscosity $\eta^{*}\left(\omega \right)$ and by the complex shear modulus $G^{*}\left(\omega \right)$ (or the compliance $J^{*}\left(\omega \right)=1/G^{*}\left(\omega \right)$) [127]:(52)$$G^{*}\left(\omega \right)=G^{{}^{\prime }}+iG^{{}^{\prime \prime }}=i\omega \eta^{*}=i\omega \left(\eta^{{}^{\prime }}+i\eta^{{}^{\prime \prime }}\right)$$
being $G^{{}^{\prime }}$ and $G^{{}^{\prime \prime }}$ the storage (or elastic) and loss moduli, respectively. In an oscillatory experiment, such as the one performed here, these moduli are obtained from the measurement of the time dependence of the stress, σ as:(53)$$\sigma =\gamma (G^{{}^{\prime }}\sin\left(\omega t\right)+G^{{}^{\prime \prime }}\cos\left(\omega t\right))$$
where γ is the strain amplitude. The equation showing that the stress amplitude varies as $\sigma =\sigma_{0}\sin\left(\omega t+\delta \right)$, also proposes the connection between the stress/strain phase angle δ(ω) and the moduli as $G^{{}^{\prime }}=\left(\sigma_{0}/\gamma \right)\cos\delta$ and $G^{{}^{\prime \prime }}=\left(\sigma_{0}/\gamma \right)\sin\delta$: so that it is: $G^{{}^{\prime \prime }}/G^{{}^{\prime }}=\tan\delta$. In addition, the shear (*G*) and elastic moduli (*E*, Young’s and *K*, bulk moduli) are related by means of the Poisson’s ratio: υ=E/2G−1 [127].

One of the early studies on the MCT viscoelasticity was developed in order to analyze the frequency dependence of $G^{{}^{\prime }}\left(\omega \right)$ and $G^{{}^{\prime \prime }}\left(\omega \right)$, of colloidal hard spheres at different concentrations (*C*) [128]. After that, this topic became a relevant subject of interest for glass forming and complex materials [53,129,130,131,132,133,134,135]. Nowadays, it represents, via the FSD crossover idea [21], the connection point between the complex fluid dynamics and the energy landscape [19,20].

Figure 18 illustrates the ω dependence of $G^{{}^{\prime }}\left(\omega \right)$ and $G^{{}^{\prime \prime }}\left(\omega \right)$ measured, in the L64/D_2_O suspensions, for φ=0.49 and 0.52, by using a strain controlled oscillating rheometer in a cone-plate geometry. The linear viscoelastic moduli were determined, in the range 0.0147<ω<9.613 s${}^{-1}$. The shear viscosity η(T,φ) was instead obtained at the fixed frequency of ω=1 s${}^{-1},$ by ensuring a linear response. The corresponding data are illustrated in Figure 19, for different φ in an Arrhenius plot.

The MCT formalism was used to explain the cage dynamics by considering hard spheres light scattering data, in terms of the β decay. An MCT feature is that near the glass concentration ($C_{g}$) the temporal autocorrelation functions of all variables coupled to density fluctuations are identical in form. Thus, it is possible to assume (Equation (Equation 24) that the generic stress autocorrelation function *S* has the same functional form as the $\phi_{q}\left(t\right)$ and by using the generic, asymptotic MCT form for the β regime on the liquid side of $C_{g}$ [31], it is:(54)$$S_{\tau \tau }\left(t\right)=f_{\tau \tau }^{C}+h_{\tau \tau }c_{\sigma }\left[{\left(t/t_{\sigma }\right)}^{-a^{\prime }}-B{\left(t/t_{\sigma }\right)}^{b^{\prime }}\right]=f_{q}+c_{\sigma }h_{q}g_{\pm }\left(t/t_{\sigma }\right)$$
including the nonergodicity parameter, $f_{\tau \tau }^{C}$, the critical amplitude and scale, $h_{\tau \tau }$, $c_{\sigma }$, and the β-scaling time $t_{\sigma }$; the constraints on the behavior of these parameters are: $f_{\tau \tau }^{C}$, and $h_{\tau \tau }$ concentration independent, a scaling for $c_{\sigma }∼\sigma^{1/2}$ and $t_{\sigma }∼\sigma^{1/(2A\overset{`}{ı})}$ with the separation parameter σ $=\left(C_{g}-C\right)/C_{g}$. These are predicted (for hard spheres) $a^{\prime }=0.301,b^{\prime }=0.545$, and β=0.963 [31].

In terms of the viscoelasticity concepts, the near-glass contribution to the complex shear modulus is $G_{g}^{*}\left(\omega \right)=G_{0}\left[i\omega S_{\tau \tau }^{*}\left(\omega \right)\right]$, with $S_{\tau \tau }^{*}\left(\omega \right)$ being the Fourier transform of the stress autocorrelation function, and $S_{0}$ the thermodynamic derivative of the stress with respect to the strain, which sets the scale of the stress relaxation. The real and imaginary parts of $S_{g}^{*}\left(\omega \right)$ contribute to the storage and loss moduli, respectively. The high-ω behavior is not described within MCT. Instead, it is possible to incorporate the energy storage effects due to Brownian motion by using a form calculated for a diffusional boundary layer [136]. In such a case, ($\omega a^{2}/D_{s}∼$$10^{1}$–$10^{2}$) can be considered only the high-ω asymptotic form, as predicted by kinetic theory [137]:(55)$$G_{D}^{\prime }\left(\omega \right)=\frac{3}{5\pi }\frac{k_{B}T}{a^{3}}C^{2}g\left(2a,C\right)$$
where $\tau_{D}=a^{2}/D_{s}$ is the diffusional time obtained by the *C*-dependent short-time $D_{s}$. For the high *C*, the radial pair distribution function at contact can be approximated by g(2a,C)=0.78/(0.64−C) (coherent with MD studies indicating a divergence at random close packing [138]). $G_{D}^{\prime }\left(\omega \right)$ reflects the additional driving force for diffusional motion that arises from the hard sphere interaction, which prevents contact when the shear brings the particles closer to each other.

Because of causality, Kramers–Kronig relations requiring a similar contribution to the loss modulus $S_{D}^{\prime \prime }\left(\omega \right)$, and the high-ω contribution to viscosity, $S_{S}^{\prime \prime }\left(\omega \right)=\eta_{\infty }\omega$ must be included. By assuming that the stress correlation functions are statistically independent, the elastic moduli are thus the sum of the individual contributions. Such that the following was proposed [128]:(56a)$$G^{\prime }\left(\omega \right)=G_{P}+G_{\sigma }\left[\Gamma \left(1-a^{\prime }\right)\cos\left(\frac{\pi a^{\prime }}{2}\right){\left(\omega \tau_{\sigma }\right)}^{a^{\prime }}-B\Gamma \left(1+b^{\prime }\right)\cos\left(\frac{\pi b^{\prime }}{2}\right){\left(\omega \tau_{\sigma }\right)}^{-b^{\prime }}\right]+G_{D}^{\prime }\left(\omega \right)$$
(56b)$$G^{\prime \prime }\left(\omega \right)=G_{\sigma }\left[\Gamma \left(1-a^{\prime }\right)\sin\left(\frac{\pi a^{\prime }}{2}\right){\left(\omega \tau_{\sigma }\right)}^{a^{\prime }}-B\Gamma \left(1+b^{\prime }\right)\sin\left(\frac{\pi b^{\prime }}{2}\right){\left(\omega \tau_{\sigma }\right)}^{-b^{\prime }}\right]+G_{D}^{\prime \prime }\left(\omega \right)+\eta_{\infty }\omega$$

The plateau modulus, $G_{P}=G_{0}f_{\tau \tau }^{C}$, contributes only to $G^{\prime }\left(\omega \right)$ and is the overall elasticity magnitude approaching the glass. Moreover, as expected from the MCT description, the storage modulus has an infection point at the plateau value, while the loss modulus has a minimum; the frequency of these is set by $1/t_{\sigma }$.

The $G_{\sigma }=G_{0}h_{\tau \tau }c_{\sigma }$ amplitude determines the degree of variation of $G^{\prime }\left(\omega \right)$ about its plateau and the $G^{\prime \prime }\left(\omega \right)$ magnitude at the minimum. By fitting both $G^{\prime }\left(\omega \right)$ and $G^{\prime \prime }\left(\omega \right)$ by means of these latter equations, this model was compared, at different concentration, with the measured data, using $G_{P}$, $G_{\sigma }$, $t_{\sigma }$, $D_{S}$ and $\eta_{\infty }$ as fitting parameters. In such a way an excellent agreement between theory and experiments was obtained; in particular, capturing correctly the plateau behavior observed in $G^{\prime }\left(\omega \right)$. Subsequently this approach was reconsidered in order to account for viscoelastic measurements in L64/D${}_{2}$O copolymer micellar system with short-range inter-micellar attractive interactions, characterized, in different regions of the composition–temperature phase diagram, by the existence of a percolation line (PT) and a kinetic glass transition (KGT) [139].

In particular, for the structural arrest at the KGT the frequency-dependent shear modulus behaviors have been investigated. As previously shown from PCS experiment, values of b=0.6 and λ=0.7 have been measured, whereas a universal plot, for different ISFs, of the von Schweidler law gives γ=2.3 [35]. Being MCT developed by using cage effects (or clustering) that determine the temporal (or frequency) dependence of relaxations [10], by considering the system structure it was assumed that the time-dependent shear viscosity can be written, in terms of the structure factor as [12]
(57)$$\eta \left(t\right)=\frac{k_{B}T}{60\pi^{2}}\int_{0}^{\infty }dq{\left[nq^{2}\frac{dlnS(q)}{dq}\varphi \left(q,t\right)\right]}^{2}$$
with the complex shear modulus $G^{*}\left(\omega \right)$ given by the corresponding Laplace transform:$$G^{*}\left(\omega \right)=i\omega \int_{0}^{\infty }dte^{i\omega }t\eta \left(t\right)$$
by assuming that the β density correlator region is a function of a scaled frequency variable $\hat{\omega }=\omega t_{\sigma }$ corresponding to the scaled time variable $\hat{t}=t/t_{\sigma }$ we can write:(58)$$G^{*}\left(\omega \right)=k_{B}T\left[-g_{0}+g_{1}c_{\sigma }\chi^{*}\left(\hat{\omega }\right)+g_{2}c_{\sigma }^{2}\psi^{*}\left(\hat{\omega }\right)\right].$$
and
(59)$$g_{0}=\frac{k_{B}T}{60\pi^{2}}\int_{0}^{\infty }dq{\left[q^{2}\frac{dlnS(q)}{dq}f\left(q\right)\right]}^{2}$$
(60)$$g_{1}=\frac{k_{B}T}{60\pi^{2}}\int_{0}^{\infty }dq{\left[q^{2}\frac{dlnS(q)}{dq}f\left(q\right)h\left(q\right)\right]}^{2}$$
(61)$$g_{2}=\frac{k_{B}T}{60\pi^{2}}\int_{0}^{\infty }dq{\left[q^{2}\frac{dlnS(q)}{dq}h\left(q\right)\right]}^{2}.$$
with the susceptibilities defined as:(62)$$\chi^{*}\left(\hat{\omega }\right)=\int_{0}^{\infty }d\hat{t}e^{i\hat{\omega }\hat{t}}g_{-}\left(\hat{t}\right);\psi^{*}\left(\hat{\omega }\right)=\int_{0}^{\infty }d\hat{t}e^{i\hat{\omega }}\hat{t}g_{-}^{2}\left(\hat{t}\right).$$
and related by the MCT equation for the β correlator as $\lambda \chi^{*}\left(\hat{\omega }\right)+\psi^{*2}\left(\hat{\omega }\right)+1=0$. Obtaining the final expressions for the storage and the loss moduli:(63)$$G^{{}^{\prime }}=-g_{0}+g_{1}c_{\sigma }\chi \left(\hat{\omega }\right)\left[1-\frac{g_{2}c_{\sigma }}{\lambda g_{1}}\frac{1+\chi^{{}^{\prime }2}\left(\hat{\omega }\right)-\chi^{\prime \prime 2}\left(\hat{\omega }\right)}{\chi^{\prime }\left(\hat{\omega }\right)}\right]$$
(64)$$G^{{}^{\prime \prime }}=g_{1}c_{\sigma }\chi^{{}^{\prime \prime }}\left(\hat{\omega }\right)\left[1-\frac{2g_{2}c_{\sigma }}{\lambda g_{1}}\chi^{\prime }\left(\hat{\omega }\right)\right]$$

When close to the KGT, and to the lowest order in the control parameter σ, $G^{{}^{\prime \prime }}$ behaves like the susceptibility showing, as proposed in Figure 18, minima and scaling properties. Being that the $G^{{}^{\prime }}$ data is not accurate to allow fitting with MCT, we considered the measured values of $G^{{}^{\prime \prime }}$ using the amplitude parameters $G_{1}=g_{1}c_{\sigma }$ and $G_{2}=2g_{2}c_{\sigma }/(\lambda g_{1}$), the frequency $\omega_{\sigma }$ and the exponent parameter λ.

For the MCT, the scaling in the β-correlation region implies the scaling of frequency and the loss modulus at the minimum. In particular, the $\omega_{min}$ position and the intensity of $G_{min}^{{}^{\prime \prime }}$ depend sensitively on the control parameter; thus, it is easy to express the two scales $1/t_{\sigma }$ and $G_{1}$ in terms of $\omega_{min}$ and $G_{min}^{{}^{\prime \prime }}$ so that we also expect $\omega_{min}\approx {\left|\sigma \right|}^{1/2a}$ and $G_{min}^{{}^{\prime \prime }}\approx {\left|\sigma \right|}^{1/2}$.

This behavior is shown in Figure 20, from the data analysis of the $\omega_{min}\left(T\right)$ and $G_{min}^{{}^{\prime \prime }}\left(T\right)$, at different φ on approaching $T_{c}$. Unfortunately, these latter scaling forms are valid only approaching the criticality of the ideal MCT and the FSC. Nevertheless, another MCT approach can be considered to describe the obtained data. A form successfully used in the analysis of $\chi^{\prime \prime }\left(\omega \right)$ data, measured by depolarized light scattering with which a MCT master curve can be found, is:(65)$$G^{{}^{\prime \prime }}\left(\omega \right)=G_{\min}^{{}^{\prime \prime }}\left[b{\left(\omega /\omega_{min}\right)}^{a}+a{\left(\omega /\omega_{min}\right)}^{b}\right]$$

This interpolation form (IF) is equivalent to the main MCT form for the β-correlators in the time regime [10]. The full curves in Figure 21 and Figure 22 are the interpolation form (IF) for λ=0.75 and γ=3.1 (the average value obtained from the viscosity fitting). In particlular, these figures report data for φ=0.50, φ=0.505 and φ=0.53, illustrating the rescaled loss moduli $G^{{}^{\prime \prime }}\left(\omega \right)/G_{\min}^{{}^{\prime \prime }}$ versus $\omega /\omega_{\min}$, in a log-log plot. Figure 21 also shows in the left panel the $G^{{}^{\prime \prime }}\left(\omega \right)$ data measured at different temperatures as a function of the frequency.

### 5.2. Percolation

As said, the percolation models applied to complex fluids predict a universal scaling behavior [52,140]. An important step to understand the corresponding viscoelastic behaviors was the recognition [7,141] that the elastic modulus of a gel and the viscosity of a sol are related to the conductance behavior of a random resistor or a random superconducting network.

According to this analogy, the complex modulus $G^{*}\left(p,\omega \right)$ is expected to behave as the AC conductivity of a resistor–capacitor random mixture (*p* is the control parameter (the cross-linker concentration, the φ of polymers molecules, *T*, etc.)). For $\varepsilon =\left|\left(p-p_{c}\right)/p_{c}\right|\ll 1$, the percolating structure is characterized by a correlation length that scales as $\xi ∼\varepsilon^{-v}$ and a relaxation time τ, scaling as $\tau \approx \xi^{z}\approx \varepsilon \bar{{}^{z}}$, where *v*, *z* and $\bar{z}=vz$ are critical exponents [29]. Customary exponents used to describe viscoelastic singularities near the threshold in the static regime are *s*, *t* and Δ [52,141], defined respectively by the following power laws:(66)$$\eta_{0}∼\varepsilon^{-s}\;\;\;\;\;\;\;\;\;\;\;\;\;\;\;\;\;\;\;\;\mathrm{and}\;\;\;\;\;\;\;\;\;\;\;\;\;\;\;\;\;\;\;\;G_{0}∼\varepsilon^{-t}$$
where $G_{0}$ is the static elastic modulus (the monomers modulus at some microscopic time scale $\tau =1/\omega_{0}$). Instead, at the threshold, $G^{{}^{\prime }}\left(\omega \right)$ and $G^{{}^{\prime \prime }}\left(\omega \right)$ have this dependence:(67)$$G^{{}^{\prime }}\left(\omega \right)∼G^{{}^{\prime \prime }}\left(\omega \right)∼\omega^{\Delta }$$

The mentioned analogy of percolating viscoelasticity with conductivity allows to postulate (for $G^{*}$ in the limit ω,ε→0) a general scaling form [141,142]. The characteristic ω-power law behavior of both $G^{{}^{\prime }}\left(\omega \right)$ and $G^{{}^{\prime \prime }}\left(\omega \right)$ for temperatures near the threshold is shown in Figure 23, for four volume fractions.

In percolating systems, the variation of the mechanical properties is correctly described as:(68)$$G^{*}\left(\omega ,\varepsilon \right)=G_{0}\varphi^{\pm }\left(i\omega /\omega_{0}\right),\;\;\;\;\;\;\;\;\;\;\;\;\;\;\;\;\;\;\;\;\;\;\;\mathrm{with}\;\;\;\;\omega_{0}\approx \xi^{-z}\approx \varepsilon \bar{{}^{z}}$$

At low frequencies $\omega \ll \omega_{0}\varepsilon^{s+t}$, for $p<p_{c}$ and $p\to p_{c}^{-}$ is:(69)$$\varphi^{-}\left(i\omega /\omega_{0}\right)=B^{-}\left(i\omega /\omega_{0}\right)+C^{-}{\left(i\omega /\omega_{0}\right)}^{2}$$
and for $p>p_{c}$ and $p\to p_{c}^{+}$:(70)$$\varphi^{+}\left(i\omega /\omega_{0}\right)=A^{+}+B^{+}\left(i\omega /\omega_{0}\right)$$
and at intermediate frequency regime, $\omega_{0}\varepsilon^{s+t}\ll \omega \ll \omega_{0}$:(71)$$\varphi^{-}\left(i\omega /\omega_{0}\right)\approx \varphi^{+}{\left(i\omega /\omega_{0}\right)}^{\Delta }$$
leading to the correct ω dependence for the elastic (viscous part) $G^{{}^{\prime }}$ ($G^{{}^{\prime \prime }}$) for ω→0. Below the threshold is $G^{{}^{\prime }}\approx \omega^{2}$, and very above $G^{{}^{\prime }}\approx const$ ($G^{{}^{\prime \prime }}\approx \omega$). With:(72)$$\eta_{0}=\underset{\omega ⟶0}{Lim}G^{*}\left(\omega \right)/\omega$$

Finally, the general scaling forms lead to the relations between the scaling exponents
(73)$$\bar{z}=s+t\;\;\;\;\;\;\;\mathrm{and}\;\;\;\;\;\Delta =t/\bar{z}=t/\left(s+t\right)$$

In addition, in the threshold regime, the frequency power-law of the complex modulus $G^{*}$ has a remarkable consequence: $G^{{}^{\prime \prime }}/G^{{}^{\prime }}$ have an universal critical value. More precisely, considering that in rheology the loss angle δ is defined just as $\tan\delta =G^{{}^{\prime \prime }}/G^{{}^{\prime }}$ one has:(74)$$\delta_{c}=\frac{\pi }{2}\Delta =\frac{\pi }{2}\frac{t}{\left(s+t\right)}$$

An MD simulation study [143] gave for the exponents *s* and *t* the following values: s=0.75±0.04, and t=1.94±0.1, so that Δ=0.72±0.04. Now we will give specific focus to the viscoelasticity to clarify some of the related properties. Figure 24 illustrates such a situation in an Arrhenius plot of the normalized viscosities. More precisely, the normalized crossover temperature is reported in the main plot, $\eta \left(T\right)/\eta \left(T_{c}\right)$ vs. $\varphi_{c}/\varphi$, for T=318, 315, 313, 311, 309 and 305 K (data also reported in Figure 18). As it can be seen two separate behaviors above and below $\varphi_{c}/\varphi =1$ are clearly shown; a precise Arrhenius behavior for $\varphi >\varphi_{c}$, and in the opposite case data follow the MCT power law $\left(\eta =\eta_{0}|\left(\varphi -\varphi_{c}\right)/\varphi_{c}{|}^{-\gamma }\right)$).

This situation gives further evidence of the universal behavior proposed for many supercooled fluids, see, e.g., Ref. [144]. The inset shows the viscosity data in a scaled log-log plot with temperature as control parameter ($\sigma =\left(T_{c}-T\right)/T_{c}$) for φ=0.25,0.4,0.4,0.49,0.5 and 0.52. In both the cases from the data fitting of the fragile region (*T* and φ as control parameters), it was found that the exponent γ ranges from approximately 2.7 to 3.1. In addition, all the calculated $T_{c}$ values are located near above the system percolation threshold (for φ≥0.49), and the fragile region extends from $T_{c}$ to the dynamic arrest.

The behavior of these data clearly shows not only the validity of the MCT power laws but also the coincidence between the corresponding threshold value with the onset of the FSC. While the data below $\varphi_{c}$ are described by the order parameter power law, above the behavior is absolutely Arrhenius, with roughly the same activation energy.

It should be emphasized that the viscosity data for lines shown for $\varphi >\varphi_{c}$ are obtained by means of the percolation power law $|\varphi_{c}{-\varphi |}^{-k}$, with k=1.3.

In the final part of our work we will consider how an MCT approach can describe the viscoelasticity of elastic material (filled rubber). In fact, we use DMA data of SBR-rubbers (Styrene Butadiene Rubber) filled with Carbon Nanotubes (CNT) at different concentration, Carbon Black (CB) and a mixture of them. A tensile strength test was selected to evaluate the complex modulus $E^{*}$. We consider the data measured in a temperature range 233<T<303 K for frequencies from 0.8 to 100 (s${}^{-1}$).

These systems behave as very highly viscous liquids and are characterized by a calorimetric glass transition and also a PT.

As said, linear response experiments are usually reported in terms of the ω dependence of the imaginary part of the studied response, the “loss”. If the shape of the loss peak in a log-log plot is *T* independent the liquid obeys to the so called *time-temperature superposition* (TTS) with respect to the response function. By considering the system average relaxation time τ(T), from the mathematical point of view, TTS is obeyed whenever *N* and ψ functions exist such that χ(ω,T)=N(T)ψ(ω,τ(T)).

When TTS applies response functions are easily determined over many frequency orders of magnitude, by combining measurements at different *T*. This procedure works even if only one or two decades of frequency are directly accessible (as, e.g., elastic moduli). For many years TTS was assumed uncritically. More recently, it has been the general opinion that the α process usually violates TTS.

Anyway, if TTS does not always work it cannot be assumed a priori, although for polymers scientists TTS is still often valid and largely used. The shift factors above $T_{g}$, obtained during the construction of TTS master curves, were successfully described by an equation equivalent to the Vogel–Fogel–Tamman (SA) form, i.e., the Williams–Landel–Ferry equation [145]:$$\log S_{F}=\log\frac{\tau (T)}{\tau (T_{ref})}=-\frac{C_{1}\left(T-T_{ref}\right)}{C_{2}+\left(T-T_{ref}\right)}$$
where $C_{1}$ and $C_{2}$ depend on the material and on $T_{ref}$. An expression that usually holds for polymers over the range $T_{g}<T<T_{g}+100\;K$ and when $T_{ref}$ is identified with Tg it was seen that $C_{1}$ and $C_{2}$ assume “universal” values close to 290.44 and 324.6 K, respectively [145].

It has been also proposed that in the less viscous regime at higher *T*, thus before $T_{c}$ where ideal MCT is valid, this model can be used for proper TTS predictions [146].

On following these suggestions we propose here a new method based on the linear response theory susceptibility spectra χ(ω) reflecting the system relaxation they have to show peaks with maxima and minima. Based on this basic concept we used the Equation (Equation 66) to fit the experimentally measured data of the SBR-CNT samples, obviously the loss modulus. We used as fitting parameters at a given temperature the exponent *a*, *b* and $E_{min}^{\prime \prime }$ to obtain $\omega_{min}$ and $E_{min}^{\prime \prime }$. The results of such a fitting are illustrated in the Figure 25 for the phrCNT (phr-parts per hundred rubber) of 1 and 40, respectively. It should be noted that the goodness of this procedure is demonstrated by the fact that the values obtained for $\omega_{min}$, at the different *T* for each concentration, follow a precise Arrhenius behavior (as well as the WLF shift factors). After which the TTS was made, in multiplicative terms, considering the shift between the values $\omega_{min}$ and $E_{min}^{\prime \prime }$, and the results obtained are illustrated in Figure 26.

The obtained data are proposed like curves with full symbols, obviously there is not a complete overlap between them because we have not considered vertical shift factors between the two concentrations.

Finally, we conclude, taking into due consideration the finding of dielectric relaxation measurements showing that the TTS is obeyed for the primary relaxation process at low temperatures for many molecular liquids close to the calorimetric glass transition and also indicating that TTS is linked to an $\omega^{-1/2}$ decay of the loss contribution, while the loss peak width is non-universal [147]. In this context, the hypothesis of considering the data shown in Figure 26 with an extra background contribution in the values of the loss modulus was done.

Assuming that its value is the asymptotic one of $E^{\prime \prime }$ at low frequencies, we subtracted it from the original data, obtaining essentially the two lines made up of empty symbols (for the two concentrations considered). We then superimposed two straight lines as a guide for the eyes. The final result of this operation is that in the log-log representation of the figure the slope of the two straight lines is about 0.5, fully confirming the suggestion of the ref. [147]. Such a result suggests, considering the many orders of magnitude in frequency with this precise scaling-law behavior, the TTS general validity.

## 6. Conclusions

Statistical mechanical foundation to estimate the properties of glass forming materials at the gelling processes is presented. In this frame we considered the centrality of the concepts related to the idea of the energy landscape and the universality of the fragile-to-strong dynamical crossover (FSC) to understand the thermodynamic transport properties of simple and complex fluids on going from the stable to dynamical arrest, crossing the metastable supercooled phase. By assuming that the FSC is due to a change in the energy configuration of the system, precisely in terms of the energy landscape, and that the corresponding characteristic temperature $T_{x}$, is identified as the critical $T_{c}$ of the ideal MCT [21], we have proposed to consider this theory as the first basis in our study. Moreover, bearing in mind that scaling concepts, scale transformations and invariance represent together with universality the conceptual background for a clear and complete description of the chemical-physic properties of complex materials and system, we have also considered as relevant for our purposes the Percolation model together with the MCT.

If, on the one hand, PT is the simplest model displaying a phase transition, on the other, the MCT represents the method of utilizing the principles of the scaling law to calculate the divergent part of the system transport coefficients arising from the interaction among different modes of excitation. In addition, common aspects in both MCT and PT were discovered readjusting them to consider the thermodynamic reality imposed by the energy landscape. In the first case, this has been accomplished by extending the idealized version [10] and considering the hopping processes due to the FSC (and the EL) [23,24,25], whereas for PT the ideas of the facilitated models were considered [26].

In such a way, the following findings were obtained: (a) an interplay between gel and glass transition in gelling systems, with the possibility to describe the sol-gel transition using MCT, and vice versa, to use PT concepts in order to understand critical phenomena and the dynamic singularities typical of MCT; (b) clarification of the universal and singular role of the FSC on approaching the dynamic arrest. Clarifying the role of this interplay (and its validity) by properly considering some experimental results was the first of the aims of this work.

In this review, we first of all considered to present the foundations of both theoretical models in a pedagogical way, considering them both in the basic and in the extended versions. We therefore first presented the corresponding schematic models and then the details of the extended ones. Explicitly, in the case of MCT we have first considered the different models, specifically models A, B, F12 and F13, and their properties have been defined. After that, a kernel extension of the memory function and then a second extended model developed in the density autocorrelation function were considered. For percolation we have instead considered the FA facilitated models and the bootstrap extension (BP).

The F13 model which has a cusp singularity characterized by a transition line from attractive glass to repulsive glass and an A3 singularity has been characterized with great attention considering a special colloidal suspension (L64/D_2_O). Such a system is characterized by a defined glass transition as well as a percolation transition in the corresponding (T,φ ) phase diagram. Another specificity is that its PT (sol/gel) line continuously evolves towards a sol, to the attractive glass transition line. For these reasons, as shown in the present work, this colloidal system has been intensively investigated by means of different experimental techniques, including scattering (light and neutrons) as well as viscoelasticity.

For the interplay between the two models, the following connections are interesting: (i) FA, BP and MCT model B (supporting the idea that the FA model in mean field reproduces entirely MCT); (ii) FA in the mean field BP with the F13 schematic model. More explicitly, in these results it is clearly proposed that: a geometrical interpretation leads to associate the BP to the discontinuous transition of model B and the random percolation to the continuous transition of model A. Therefore, the same general formalism can treat the transition from a liquid to an arrested state, whether glass or gel. In addition, phenomenology of some polymer systems, associated to the interplay between chemical gels (with permanent bonds) and glasses, can be interpreted in terms of the F13 MCT model, clarifying also the origin of logarithmic decays observed in the correlation function measured in these systems. At the end of this part, it has been shown how specific attractive polymer colloids can be the model systems to study the specific properties of these interplays. The absolute experimental consistency of the two MCT models extended for hopping processes was then shown not only in terms of density–density correlation function but also of response functions such as dielectric susceptibility.

In the second part, considering the validity of the two theories in terms of response functions, we explicitly studied the viscoelasticity of some systems in terms of both MCT and PT. In particular, we have explicitly considered the dependencies of the loss and shear moduli as a function of frequency, temperature and volume fraction for the L64/D_2_O system.

Therefore, both the general methodologies and the related formalism, essentially made up of scaling laws, were proposed through a comparison with a considerable number of experimental results. Finally, we proposed a method by which the MCT formalism for viscoelasticity can be used in order to understand the properties of elastic materials by showing, as an example, the way in which the so-called time–temperature superposition (TTS) mechanism can be taken into account. Although TTS is the basis of many technological applications, it is instead usually used in an empirical way. All of this represents the current state of the art regarding the dynamic arrest. On these basis, we can certainly assert that the sol-gel and the glass transitions seem to have common kinetic mechanisms. In any case, the fundamental role of the fragile to strong dynamic crossover emerges from the proposed discussion. Its presence and characteristics have allowed the optimal reformulation of MCT from the ideal to the extended one also discovering that $T_{x}$ (of FSC) is the critical $T_{c}$. In our opinion, just the universality of both FSC and potential energy landscape represent the datum point in the perspectives of the research field of the complex materials. The ascertained interplay between PT and MCT will be equally useful, which represents the main objective of the present work. We are confident that FSC, energy landscape and PT-MCT interplay will be three elements characterizing the results of future research in material science (and related technological applications) and in particular on those of biological interest, including proteins, amyloidogenic species or tissue cells.

## Figures and Tables

**Figure 1 ijms-23-05316-f001:**
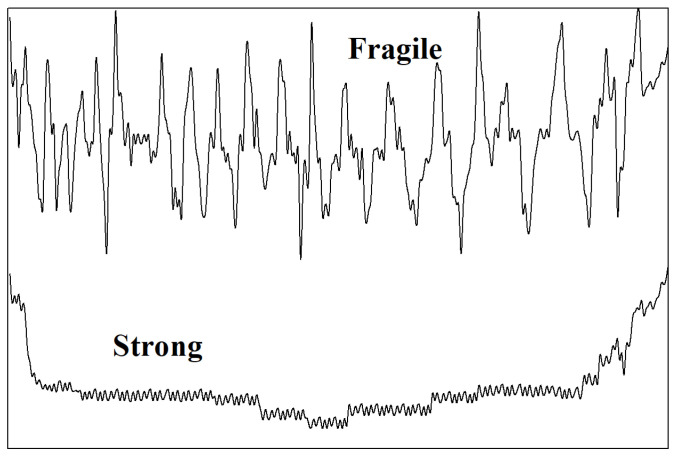
Schematic representation of the potential-energy profiles for strong Arrhenius and fragile Super-Arrhenius liquids [17].

**Figure 2 ijms-23-05316-f002:**
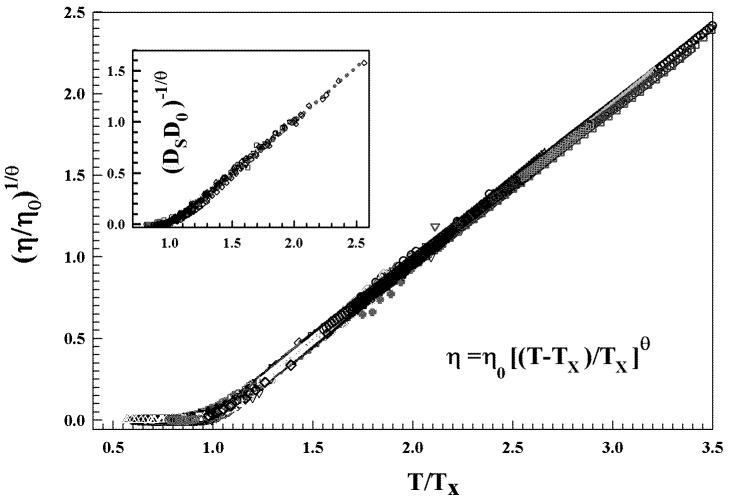
This figure illustrates the fragile-to-strong dynamical process by considering the viscosity and diffusion data of more than 80 glass forming liquids [21]. Figure adapted from Ref. [21], Copyright (2010) National Academy of Sciences.

**Figure 3 ijms-23-05316-f003:**
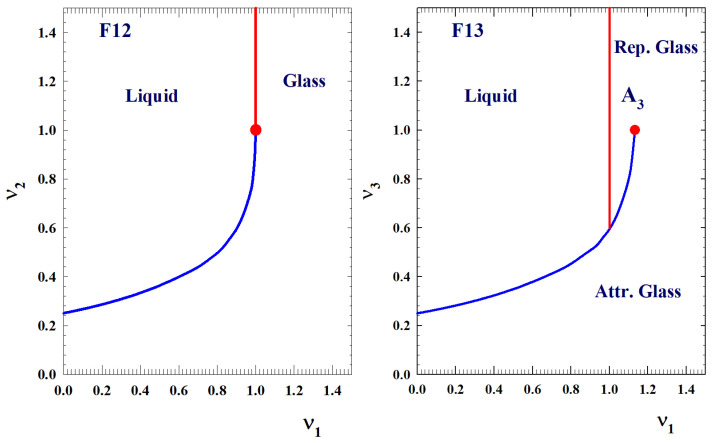
The phase diagram correspond to the F12 ($v_{1},v_{2}$-plane) and F13 ($v_{1},v_{3}$-plane) MCT models. In the latter case, the presence of cusp-like singularity A${}_{3}$ and the re-entrant behavior attractive repulsive glass are highlighted.

**Figure 4 ijms-23-05316-f004:**
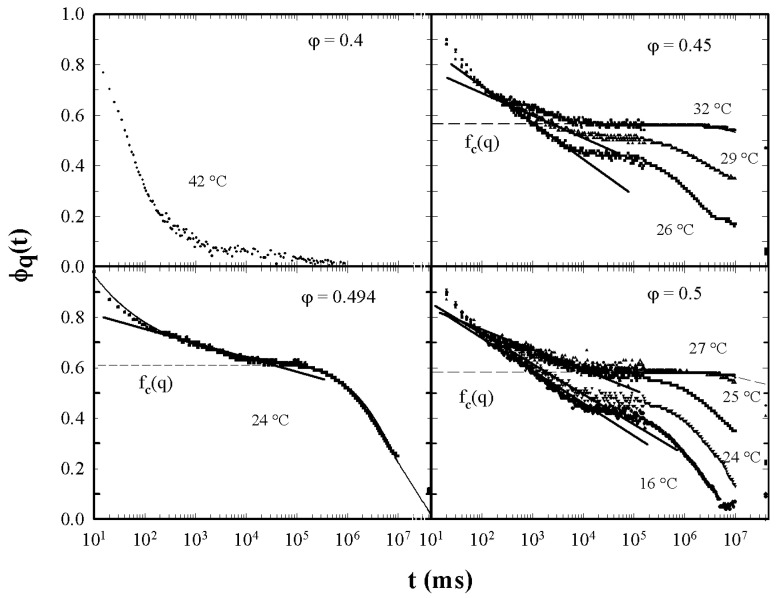
The logarithmic decays observed the measured self Intermediate Scattering Function of an adhesive colloidal system [37].

**Figure 5 ijms-23-05316-f005:**
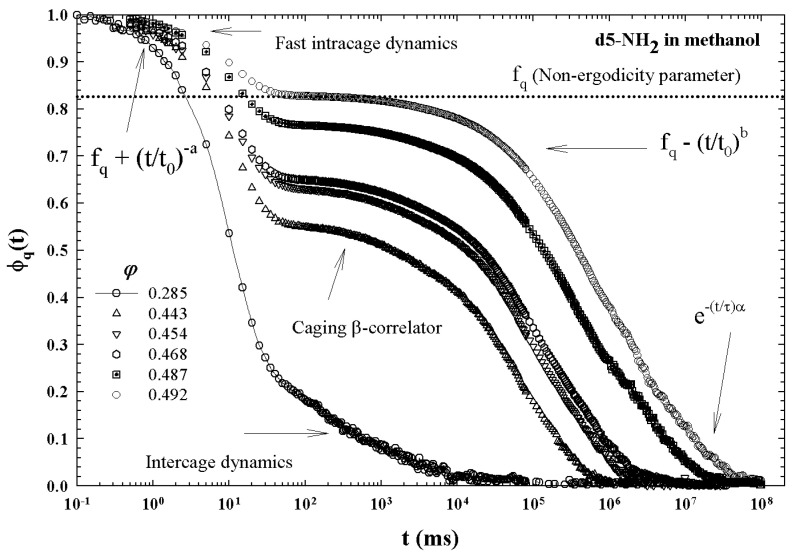
The decays of the density correlators in a supercooled liquids (colloidal suspension of a dendrimer in methanol) close to the dynamical arrest. Highlighted in particular are the different MCT regions. The initial non-universal intra-cage motion is followed by a region characterized by a power law caging dynamics. After than, there is the onset of the slow inter-cage dynamics (described by the stretched exponential).

**Figure 6 ijms-23-05316-f006:**
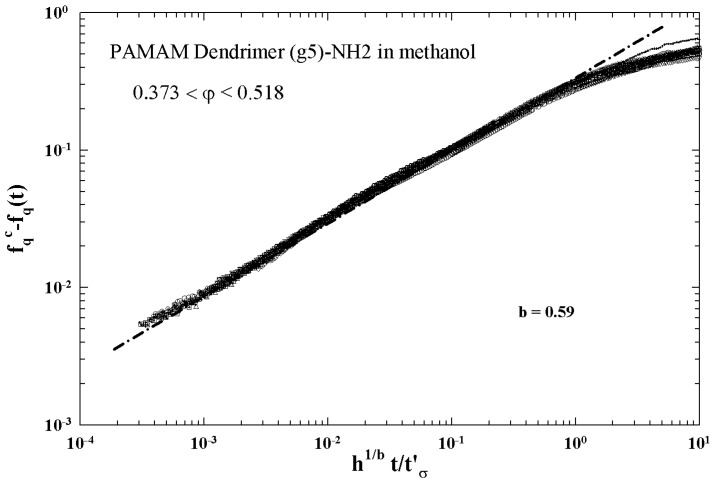
The von Schweidler universal plot measured in the previous colloidal PAMAM dendrimers suspensions.

**Figure 7 ijms-23-05316-f007:**
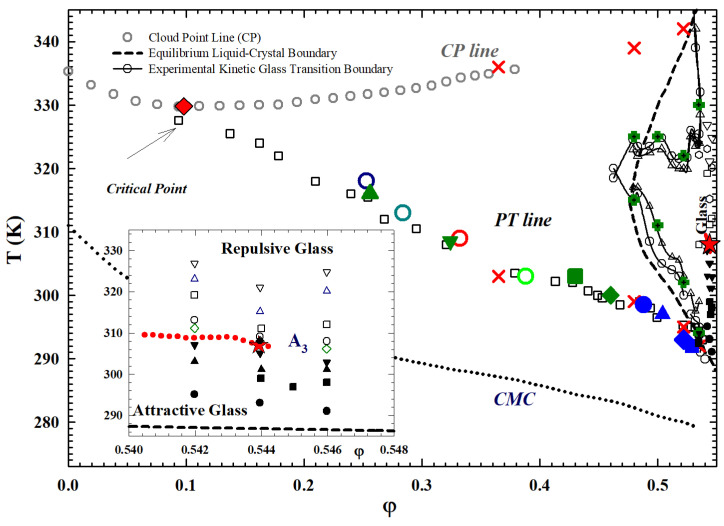
The experimental phase diagram (Temperature–Volume fraction) of the Pluronic L64 in D${}_{2}$O characterized by a critical point (cloud point line, CP), the critical micellar concentration line (CMC), and a percolation line PT [35,37,48,49]. The inset reports an expanded region of the corresponding glass phase showing the attractive-glass–repulsive-glass transition line and the cusp-like singularity A${}_{3}$.

**Figure 8 ijms-23-05316-f008:**
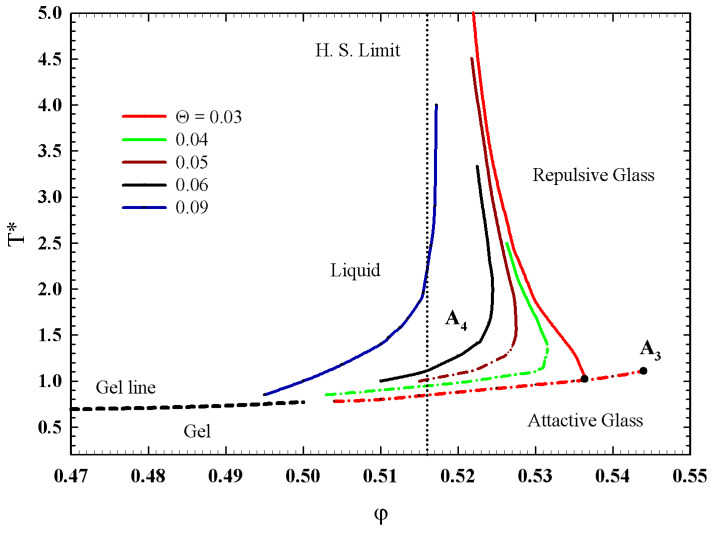
The phase diagram corresponding to a square well system described in the text in the plane $T^{*}$-φ. Reported are curves for some different fractional attractive width Θ. Can be observed here are the glass lines’ re-entrance and that the glass–glass line appears for Θ<0.04, and increases as Θ decreases. A${}_{4}$ identifies the place where the two characteristic lines (attractive and repulsive) meet continuously, while A${}_{3}$ is the end point of the attractive line. It is shown that this last line tends, at the low φ values, with continuity towards the PT line [34]. Copyright 2000 by the American Physical Society.

**Figure 9 ijms-23-05316-f009:**
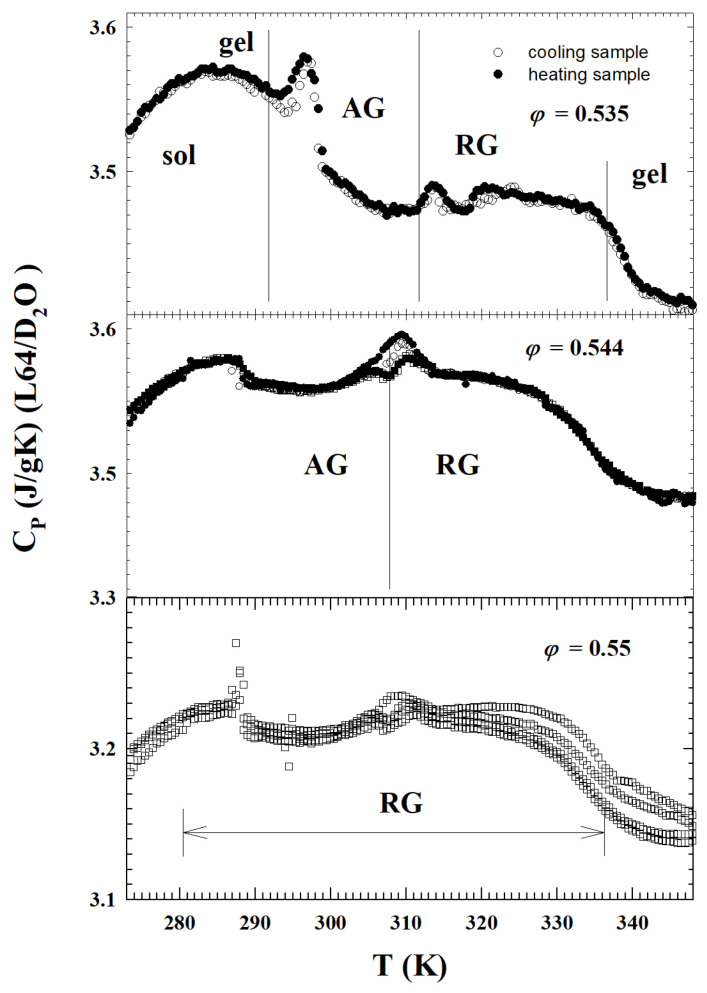
The specific heat data, Cp(J/gK) versus T, in the temperature interval 273–348 K, for our AHS system at the volume fraction φ = 0.535 (top panel), φ = 0.544 (central panel) and φ = 0.55 (bottom panel).

**Figure 10 ijms-23-05316-f010:**
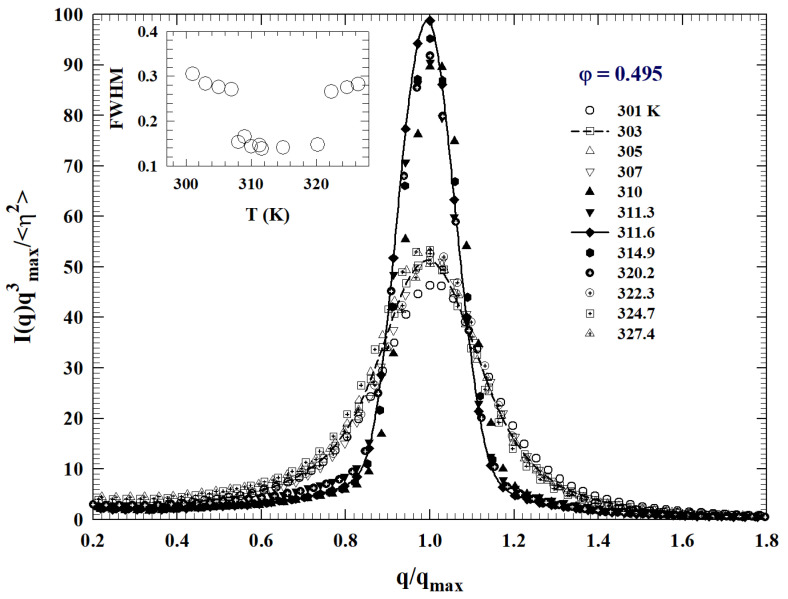
The scaling plots of the SANS intensities for φ=0.495 measured in the PluronicL64-D${}_{2}$O, at different temperatures in the range 310–327.4 K.

**Figure 11 ijms-23-05316-f011:**
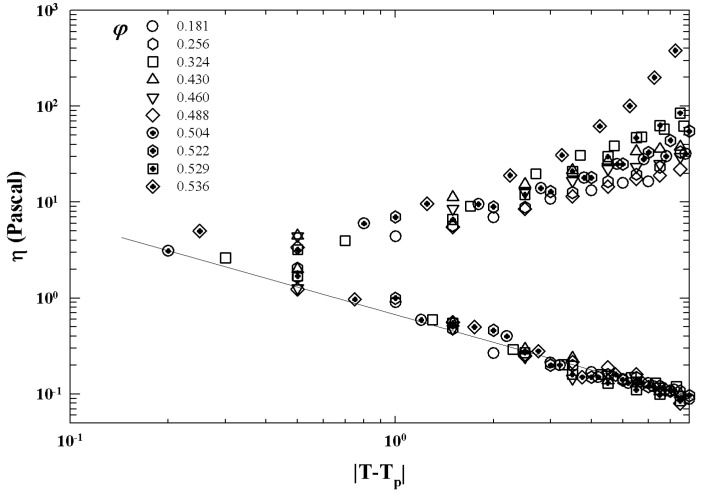
The shear viscosity η(T) measured at different volume franction reported in a log-log plot as a function of $|T-T_{p}|$.

**Figure 12 ijms-23-05316-f012:**
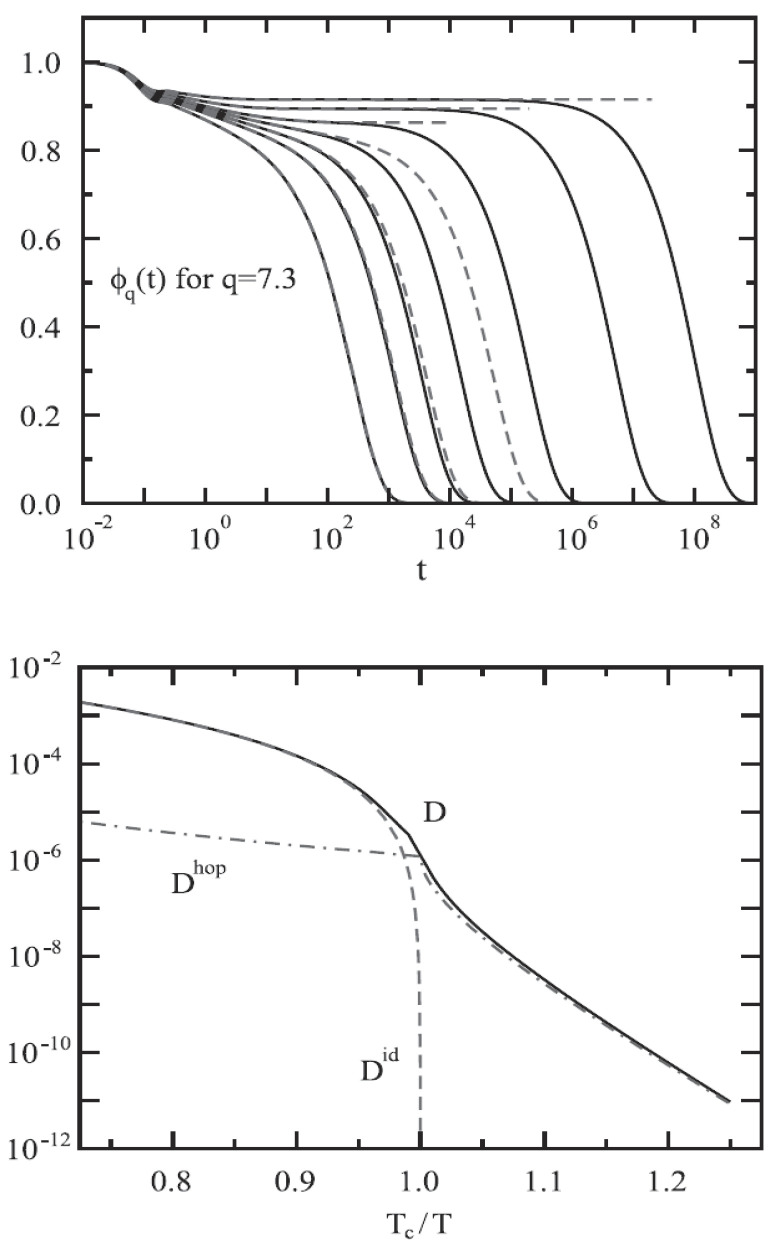
The density correlators $\phi_{q}\left(t\right)$ calculated at the peak position (q=7.3) of the static structure factor S(q) for different *T* (**top**). The dashed curves deal with the ideal MCT and shows the bifurcation (in the long *t* limit above and below $T_{c}$). The system self-diffusion *D* versus $T_{c}/T$ (**bottom**). The solid curves deal with the extended MCT while the dashed one refers to $D^{id}$ from the ideal MCT. The dash-dotted curve represents $D^{hop}$ originated to the hopping processes [67]. Adapted with permission from Ref. [67]. Copyright IOP Publishing.

**Figure 13 ijms-23-05316-f013:**
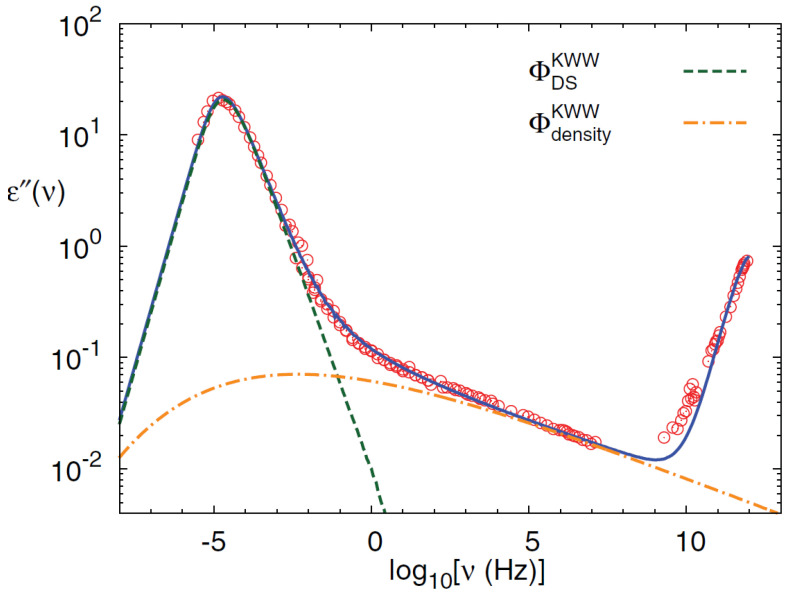
Fit of the propylene carbonate DS data (T = 157 K) by means of the extended schematic-MCT model incorporating KWW-like hopping rates (solid line). The hopping contributions stemming from the collective correlator ϕ(t) and from the probe variable correlator $\phi_{A}^{s}\left(t\right)$ are indicated separately (dashed and dash-dotted lines) [72]. Reprinted with permission from Domschke, M. et al. Phys. Rev. E 84, 627 031506 (2011). Copyright (2011) by the American Physical Society.

**Figure 14 ijms-23-05316-f014:**
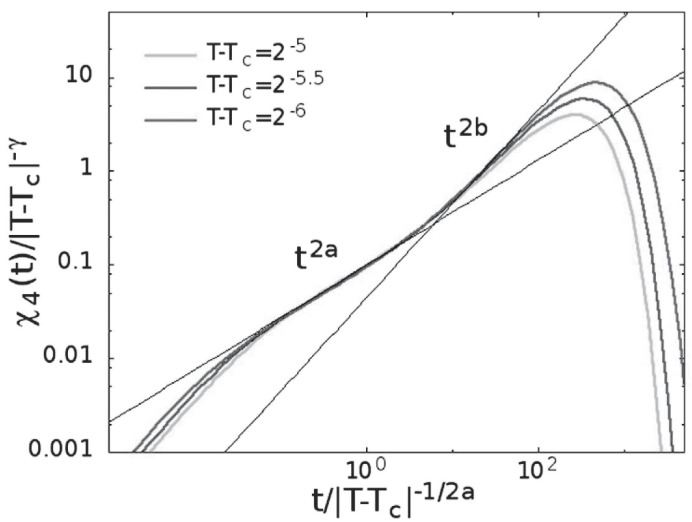
The data collapse in the β regime $t\ll t_{\alpha }$ of the dynamical susceptibility, $\chi_{4}\left(t\right)$ (for the FA model on the Bethe lattice). In particular, the scaling relation are shown: γ=1, $\tau_{\beta }∼\varepsilon^{-1/2a}$. Straight lines show the power law behaviors in the early $t^{2a}$ and late β regime ($t^{2b}$), with a=0.29 and b=0.50 [47]. Figure adapted under the terms of the Creative Commons CC BY license from Ref. [47].

**Figure 15 ijms-23-05316-f015:**
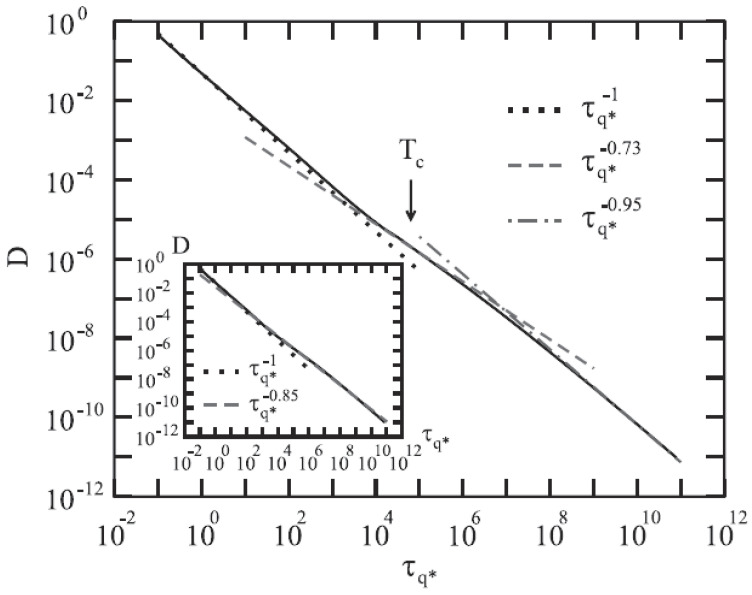
A log-log representation of the self-diffusion *D* versus α-relaxation time $\tau_{q^{*}}$ of the coherent $\phi_{q}\left(t\right)$ at the S(q) peak $q^{*}$ (solid curve). The arrow indicates $T_{c}$. The dotted line refers to the SER prediction ($D_{s}∼{\left(\tau_{q}^{*}\right)}^{-1}$), while the dashed (${\left(\tau_{q}^{*}\right)}^{-0.73}$) and dash-dotted (${\left(\tau_{q}^{*}\right)}^{-0.95}$) curves to fractional relations (see the text). Inset: *D* versus $\tau_{q^{*}}$ from the main panel (solid curve) is compared with the fractional relation (${\left(\tau_{q}^{*}\right)}^{-0.85}$ ) (dashed curve) [25]. Adapted with permission from Ref. [67]. Copyright IOP Publishing.

**Figure 16 ijms-23-05316-f016:**
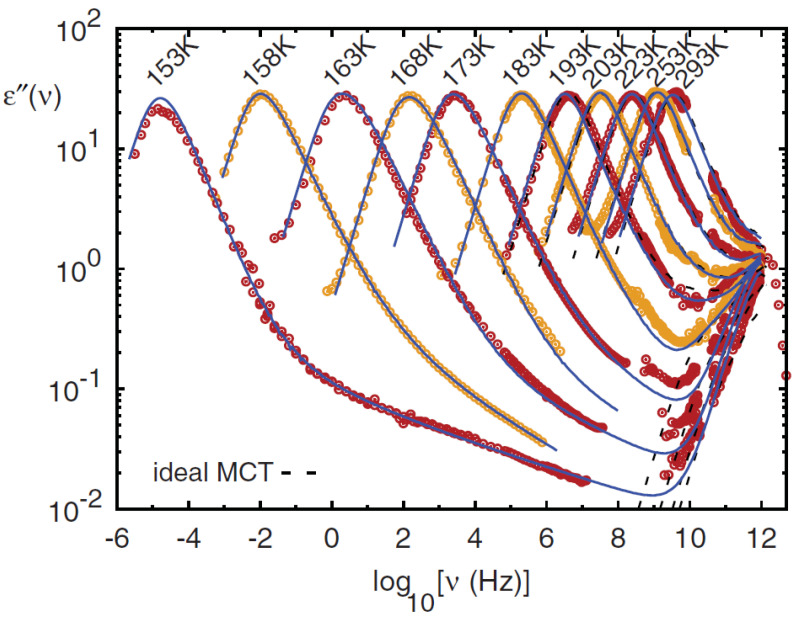
Propylene carbonate data (DS and light spectroscopy) and the related fits obtained by means of memory kernel schematic model are shown in a log-log plot. Dashed lines are fits with the ideal MCT model, i.e., without hopping term for comparison. Solid lines represent instead the fits with the extended schematic-MCT model [72]. Reprinted with permission from Domschke, M. et al. Phys. Rev. E 84, 627 031506 (2011). Copyright (2011) by the American Physical Society.

**Figure 17 ijms-23-05316-f017:**
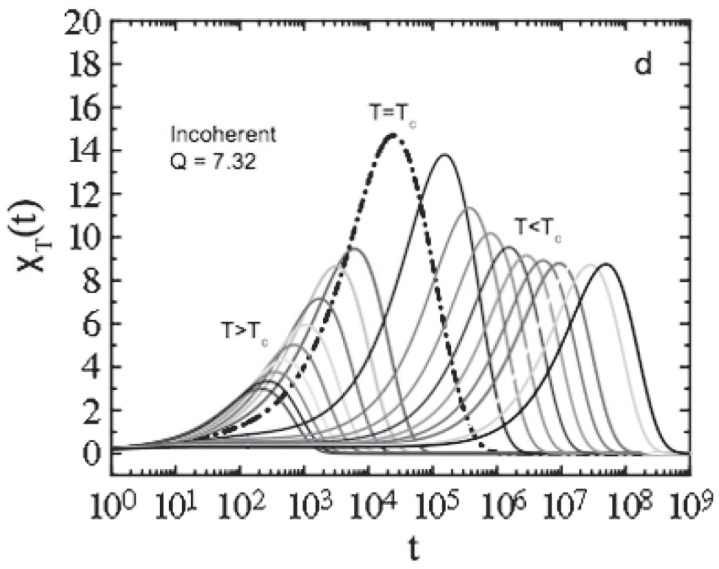
The dynamic response function χ(q,t) evaluated for a Lennard–Jones system at peak position of S(q) (q=7.3). As it can be observed, the peak height of χ(q,t), denoted as $\chi^{*}\left(q,t\right)$ increases as one approaches $T_{c}$, but below $T_{c}$ it decreases. The latter behavior is due to the dynamical crossover [121]. Adapted with permission from Ref. [68]. Copyright IOP Publishing.

**Figure 18 ijms-23-05316-f018:**
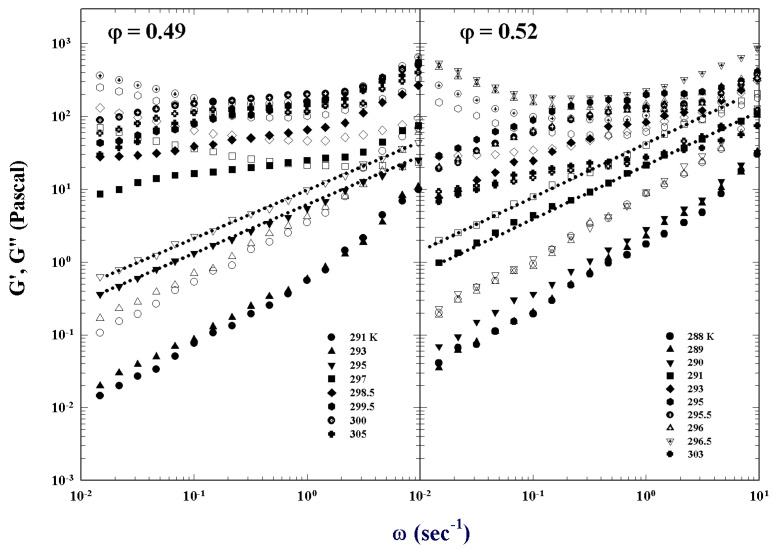
The real and imaginary part of the shear moduli for the volume franction 0.49 and 0.52. The G′ and G″ in the temperature range 291<T<305 K are reported as a function of the frequency range 0.0147<ω<9.613 s${}^{-1}$.

**Figure 19 ijms-23-05316-f019:**
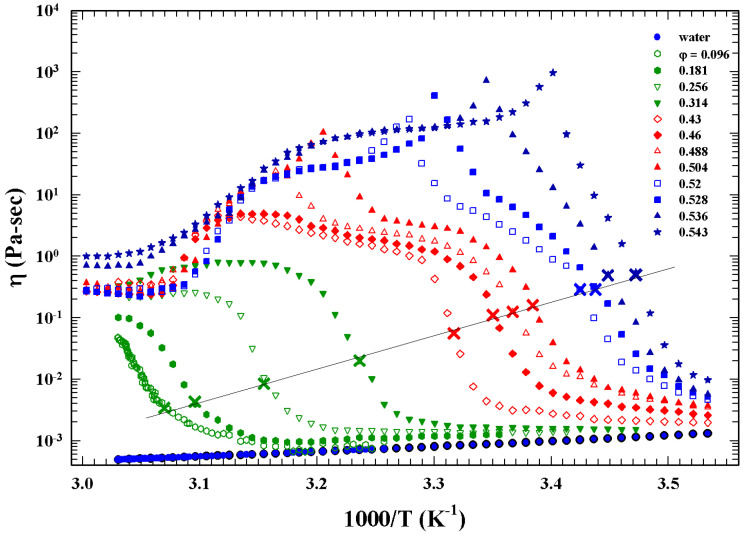
The Arrhenius plot of the shear viscosity L64/D${}_{2}$O at twelve different volume fractions as a function of 1000/T. Starting from low T, η increases steeply, first going through a PT, then to a liquid–glass transition, and finally for φ>0.53 to a glass–glass transition. Note that, for φ=0.3, only the percolation is present, denoted by crosses and straight line.

**Figure 20 ijms-23-05316-f020:**
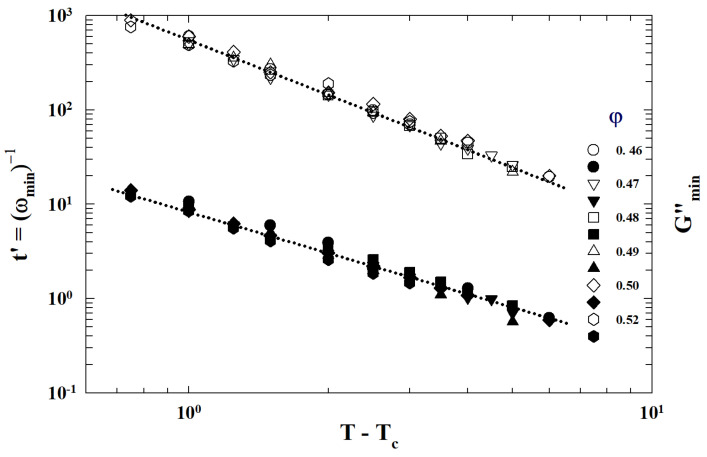
The power-law dependence of $t^{\prime }={\left(\omega_{min}\right)}^{-1}$ and $G_{\min}^{\prime \prime }$ on $|T-T_{c}|$ for some different micellar volume fractions.

**Figure 21 ijms-23-05316-f021:**
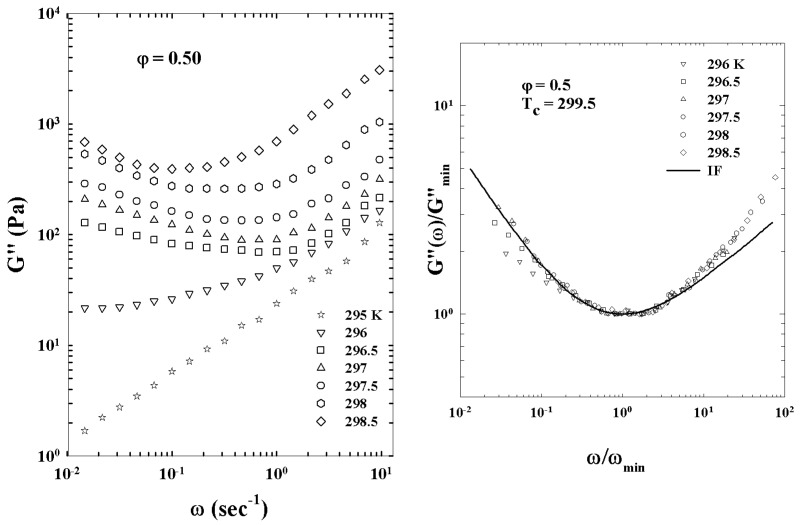
The left panel shows the $G^{{}^{\prime \prime }}\left(\omega \right)$ data measured at different temperatures for φ=0.50. From these data, the minima and their frequencies, on approaching $T_{c}$, are well evident. On the right side, the corresponding data fitting according to the MCT interpolation form (IF) are shown.

**Figure 22 ijms-23-05316-f022:**
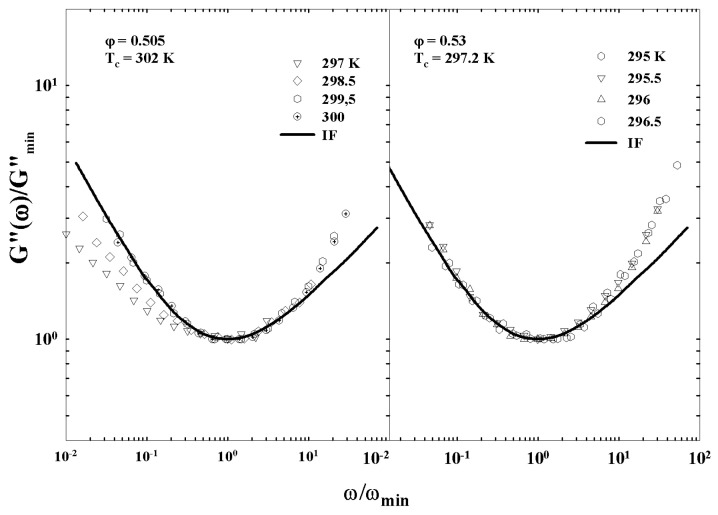
The rescaled loss moduli $G\left(\omega \right)/G_{\min}$ versus $\omega /\omega_{\min}$, for the volume fractions 0.505 and 0.53. The line is the interpolation MCT form for λ=0.75 and γ=3.1.

**Figure 23 ijms-23-05316-f023:**
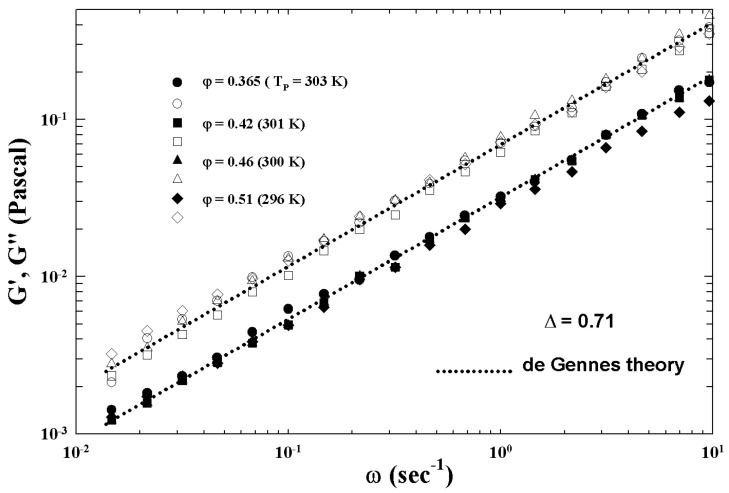
The ω-power law dependence of both $G^{{}^{\prime }}\left(\omega \right)$ and $G^{{}^{\prime \prime }}\left(\omega \right)$, near the the percolation threshold, for φ=0.355, 0.42, 0.46 and 0.51.

**Figure 24 ijms-23-05316-f024:**
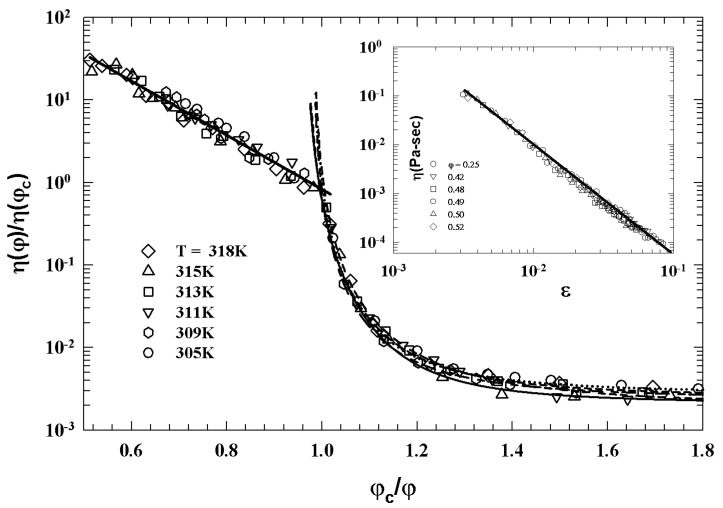
The normalized viscosity with the volume fraction as control parameter: $\eta \left(\varphi \right)/\eta \left(\varphi_{c}\right)$ vs. $\varphi_{c}/\varphi$. Data for T=305, 309, 311, 313, 315, and 318 K are reported.

**Figure 25 ijms-23-05316-f025:**
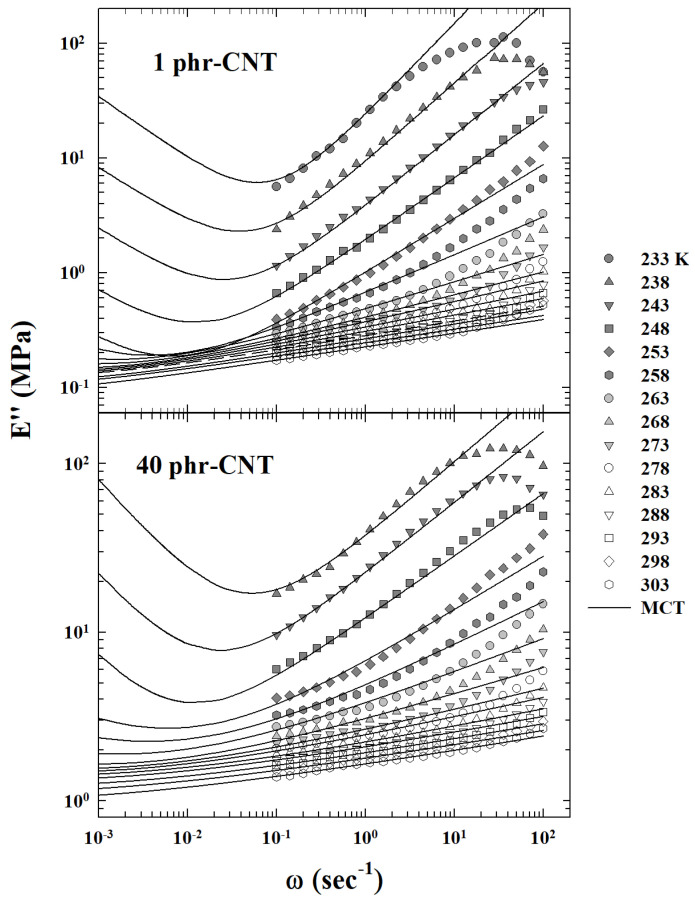
The elastic loss modulus $E^{\prime \prime }$ measured in SBR-CNT samples at the phrCNT of 1 and 40 (for different temperatures from 233 to 303 K (Symbols)). Curves are the results of the fitting procedure according to the MCT.

**Figure 26 ijms-23-05316-f026:**
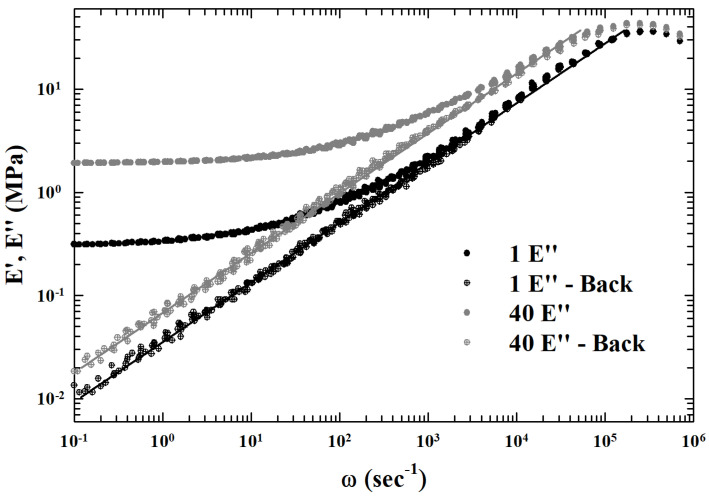
The *time–temperature superposition* (TTS) of the data illustrated in Figure 25. The curves report data (fully symbols) obtained according to the described procedure. The lines (proposed by the empty symbols) are obtained by a background subtraction. Empty symbols lines and the superimposed straight lines have an $\omega^{-1/2}$ decay, which confirms the suggested generality of TTS.

## Data Availability

The data presented in this study and not reported in tables are available on request from the corresponding author.

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
