# Peer review of "The Interplay between the Theories of Mode Coupling and of Percolation Transition in Attractive Colloidal Systems"

_ijms, 2022, doi:10.3390/ijms23105316_

Round 1

Reviewer 1 Report

The analyzed article is an important pole of understanding of the phenomena and theories related to mode coupling theory and the percolation transitions applied to colloidal systems. It is very well documented and with an impressive mathematical apparatus for those interested and experts in the field.
From the analysis of this review, I can highlight the following aspects:

  1. An extensive and in-depth revision of the English language is needed.
  2. Even in the title there are typos (theory for theory); suggested ...systems in place of "system". The authors are asked to review the entire material.
  3. It is necessary to uniformize the iconographic material (graphs or diagrams) from the point of view of the resolution.
  4. I suggest a synthetic table with some important applications of these theories to certain colloidal systems and related bibliographic references. This would add value to the article.
  5. Revise mathematical equations according to the requirements for authors.
  6. Revisiting the extensive list of bibliographic references so that it complies exactly with the editing requirements.

Author Response

We would like to thank the referee for his precise observations in order to improve the manuscript. We have carried out an accurate and careful revision of the English language; moreover, we revised the mathematical equations, the bibliography and the iconographic material in accordance with the editing requirements, accordingly to reviewer suggestions. We are sorry but unfortunately due to the shortness of the manuscript revision times we have not been able to produce satisfactorily the suggested synthetic table.

Reviewer 2 Report

Comments on the Manuscript “The interplay between the mode coupling thory and the percolation transitions in attractive colloidal system” referenced by ijms-1696932

This review reports on experimental and theoretical results (including MD simulations) of recent studies on colloidal and polymer systems. Thus, based on my comments presented below I consider that this manuscript can be accepted after Minor revision.

Title:

  • “thory” should be written “theory”

Abstract:

  • Need to be more focused on the data presented in this review. Please avoid repetition. Please clarify if this manuscript is a review or a full length article.
  • Lines 5,6: “contributed to his understanding.” – should be deleted or rephrased.

Introduction:

  • Not clear: “Together with scaling, the universality constitutes, according to the renormalization group theory, much of the correct representation of how these systems, made up of interacting subunits, behave [1].” - should be rephrased.
  • The introduction part should be thoroughly checked. There are a lot of English language mistakes. Please avoid repetition.
  • It is not clear which kind of data contains this manuscript. The authors should clarify if they discuss the already published data in the field or present new ones. So, the following sentence “Here we consider both the theories, in particular MCT, to explains some relevant and universal aspects of supercooled liquids before and at the dynamic arrest” should be rephrased. I suggest to the authors to focus their paper on one direction, afterwards it is difficult to be followed.
  • A lot of sentence without any sense, for instance “However, today and in spite of the interest, both theoretical and experimental, placed on this question in recent years, we are far from an exhaustive understanding of it; all of this, despite the enormous literature devoted.” It seems Google translate English. It is really hard to understand some part of this paper due to English language mistakes.
  • As far as there is no section 3.1.2., the section 3.1.1 should be integrated in section 3.1.

Conclusions

The conclusions in their current state are a summary of the results. A conclusion should, however, summarize the results and set them into perspective to the objectives formulated in the introduction. Moreover it has to give an outlook on the importance of the findings. So, the authors should reword/amend the conclusions accordingly. Maybe a perspective section should be included.

Finally, I consider that this manuscript presents interesting and valuable data for the scientific audience, and it would be a pity if unclear sentences will diminish its value.

Author Response

We thank the referee for his suggestions and accordingly, we proceeded to an accurate and coherent revision of the work in all its parts. We have revised the English language avoiding errors and repetition. Finally, we have added a short paragraph highlighting some possible perspectives based on the related findings in the conclusions section.